# A deep single cell mass cytometry approach to capture canonical and noncanonical cell cycle states

Meelad Amouzgar [1,2], Patricia Favaro[1], Daniel Ho[1], Trevor Bruce[1] & Sean C. Bendall [1] ✉

The cell cycle (CC) underpins diverse cell processes like cell differentiation, cell expansion, and tumorigenesis but current single-cell (sc) strategies study CC as: coarse phases, rely on transcriptomic signatures, use imaging modalities limited to adherent cells, or lack high-throughput multiplexing. To solve this, we develop an expanded, Mass Cytometry (MC) approach with 48 CC-related molecules that deeply phenotypes the diversity of scCC states. Using Cytometry by Time of Flight, we quantify scCC states across suspension and adherent cell lines, and stimulated primary human T cells. Our approach captures the diversity of scCC states, including atypical CC states beyond canonical definitions. Pharmacologically-induced CC arrest reveals that perturbations exacerbate noncanonical states and induce previously unobserved states. Notably, primary cells escaping CC inhibition demonstrated aberrant CC states compared to untreated cells. Our approach enables deeper phenotyping of CC biology that generalizes to diverse cell systems with simultaneous multiplexing and integration with MC platforms.

Regulated cell division via the cell cycle (CC) is a pervasive and ubiquitous biological process in both health and disease[1]. Here, the cellular decision to proliferate is under tight molecular control to maintain organismal homeostasis and rapidly expand immune cells while preventing unregulated expansion[2]. A central challenge of quantifying and assigning CC states is that cycling cells can be rare and heterogeneous—thus single-cell approaches are often employed to quantify CC states. The detection of proliferative signatures is further convoluted by differences in cell types, lineages, disease, and drug treatment, where intrinsic differences to CC architecture or extrinsic disruptions to CC molecular state may not be captured by low-dimensional measurement strategies[3].

Fluorescence flow cytometry can discriminate the four major CC phases of G0, G1, S, G2, and M using measurements of DNA content (i.e., PI, 7AAD), nucleotide incorporation (i.e., BrdU, EdU), protein expression and phosphorylation (i.e., cyclin B, phospho-H3)[4,5]. The advent of next-generation single-cell (sc) approaches, such as

Cytometry by Time of Flight (CyTOF) mass cytometry (MC), has enabled quantitative multiplexing beyond 45 molecular targets per cell, facilitating both regulatory molecular measurements and deep phenotyping of cell identity[6,7]. Our prior work using MC includes a combination of CyclinB1, phosphorylated-Histone H3 (pHH3, Ser28), and incorporation of 5-iodo-2′-deoxyuridine (IdU), allowing the manual annotation of G0/G1, S, G2, and M phases[8]. More granular separation of CC phases has been further achieved with the addition of phosphorylated retinoblastoma (pRb), Geminin, CDT1, and PLK1 to discriminate between G0 and G1, and split G1 and G2 phases into early and late stages[9,10].

Despite these advances in CC phase discretization, analysis methods remain overly manual, relying on gating strategies to categorize and quantify cell bins. This approach fails to recognize the continuous dynamics of CC progression, or that cells can occupy unique normal and dysfunctional CC states depending on intrinsic cell processes as well as extrinsic perturbations like drug inhibitors that

[1]Department of Pathology, Stanford University, Stanford, CA, USA. [2]Immunology Graduate Program, Stanford University, Stanford, CA, USA.
✉e-mail: bendall@stanford.edu

induce CC arrest via mechanisms such as CDK inhibition, the spindle checkpoint, or p53 activation[11]. While such advancements in quantifying CC aberrancy have been approached with microscopy to describe the molecular architecture during hypomitogenic and replication stress, an advantage of next-generation single-cell (sc) approaches, like MC, is they are well-suited for economical, high-throughput analysis of dissociated cell systems in combination with barcoding strategies for multiplexing experimental samples, enabling deeper (i.e., higher throughput and multiplexing) characterization of CC states beyond manual gating[12–15]. The global dynamics of molecules regulating CC are often conserved across different cell systems, but the exact abundance and patterning of molecules required for CC progression can differ depending on factors like cell size, genome size, replication speed, cell line origins, and cell extrinsic molecular factors[16–18]. Furthermore, disease and perturbation can disrupt the molecular patterns that define canonical CC, inducing noncanonical CC states like CyclinD1 loss in G2-phase, relicensing of DNA replication by CDT1 overexpression during G2-phase, and tetraploidy[12,19–21]. High-throughput, low-dimensional strategies may fail to characterize both canonical and noncanonical CC states.

The high-throughput, parallel quantification capabilities of scMC have allowed phenotyping of molecular states during complex biological processes in primary cells, such as TCR stimulation and T cell expansion, as well as chromatin modifications across immune cells[22,23]. To better understand the diversity of scCC states using MC, we further multiplex existing MC panels to expand CC-related measurements. Using this, we characterize the diversity of CC states across five cell lines commonly used in the literature, as well as stimulated and expanded primary human T cells. We further study the consequences of CC perturbation on scCC states using DNA synthesis inhibitors, microtubule destabilizing agents, and CDK4/6 inhibition in Jurkat cells, as well as a breadth of CDK inhibitors in primary human T cells, to quantify the patterns of scCC aberrancies in arrested and escaped cells. Using our scCC platform in primary human T cells, we reveal the CC state similarities in drug perturbation between different CDK inhibitors, the consequences of CDK inhibition on the diversity of scCC states, and expose that cells escaping CC arrest occupy aberrant CC states not observed canonically. This single-cell assay serves as an additional MC tool that can be integrated with other MC platforms like scMEP for metabolism, EpiTOF for chromatin marks, cell phenotyping, cell differentiation, hematologic cancer cells, and other measurements to tease the pervasive crosstalk between CC and other systems[22–26].

## Results

### A single-cell module for deep molecular typing of CC biology
The CC is a highly conserved biological process with tightly regulated checkpoints finely orchestrated by CC protein abundance and post-translational modifications (PTMs). This molecular machinery ensures accurate DNA replication and chromosome segregation during cell division and prevents premature S- and M-phase entry from gap phases[27]. To date, CC quantification by single-cell analysis has focused on quantification of cells residing in individual phases using landmark reporters[9]. However, recent studies demonstrate that cells can occupy diverse CC states captured with multivariate molecular signatures[12]. For example, gold-standard markers of proliferation, Ki67 and phosphorylated Rb, are dynamically expressed and can be lowly abundant in cells with other cycling signatures, like active DNA replication (S-phase) defined by IdU labeling or high cyclin B abundance (G2-phase) (Fig. 1A)[28]. For this reason, low-dimensional landmark reporters fail to capture more granular CC states in exact phases and may not accurately quantify the molecular effects of CC perturbation. To establish a solution, we created an expanded, metal-tagged antibody approach that quantifies the abundance of an increasingly diverse number of CC-related molecules including: cyclins that promote CC progression, CC checkpoint inhibitors, DNA licensing factors, PTMs of proteins that

control proliferation and mitotic entry, nucleotide incorporation during S-phase, DNA content, chromatin state, and CC regulators (Supplementary Data 1). Our scCC approach includes three sets of molecular features: (i) "minimal" includes protein and phospho-protein targets that directly control CC checkpoints and progression, (ii) "core" includes the minimal CC molecular targets paired with measurements of DNA content and replication such as DNA intercalators and IdU incorporation, and (iii) "complete" includes a wider array of measured CC-related molecules including transcription factors, chromatin state, and other CC regulators (Fig. 1B). These targets were categorized into each panel based on both variance associated with separating CC phases and functional characteristics reported in the literature (Fig. 1B, Supplementary Fig. 1A–C). This panel can be further customized to include new targets. We additionally recommend a combined experimental and in silico cleaning strategy for debris, doublet, and pre-apoptotic cell elimination that generalizes to any experimental system using CyTOF (Fig. 1C, D). With our scCC platform, we further characterize CC heterogeneity in diverse suspension and adherent cell lines (Fig. 1E–G). We establish a core CC molecular panel with breadth to directly measure diverse CC states for combinatorial sc systems analysis with other MC panels (Fig. 1E). We then extend our platform to multiplexed perturbation experiments and quantify the sc consequences of a variety of CC inhibitors in both cell lines and primary human T cells (Fig. 1F). Using a graph connectivity approach to quantify cell diversity, our platform for deeper CC profiling captures increasingly more diverse CC states as a function of additional CC-specific features across all cell lines (Fig. 1G–I, see "Methods"). Notably, there were cell-line-dependent differences in CC diversity captured by the panel in both the average diversity as well as the increase in diversity with additional features.

### Dimensionality reduction reliably captures cell cycle phase
To understand the CC molecular expression across different cell systems, we applied our molecular panel to a variety of adherent and suspension cell line types: NALM6, U937, HEL, 293T, and JURKAT cell lines, as well as primary human T cells. Given that different cell systems can have variable CC dynamics, we first sought to evaluate the diversity of CC states in these cell lines (Fig. 2A, B)[29]. We leverage multiplexed palladium barcoding to eliminate procedural variation between samples and reduce doublets, simultaneously process and stain cells for fair molecular comparisons across cell lines, and perform live singlet acquisition using event length, barium, DNA, and standard CC gating practices to remove doublets, debris, dead, and pre-apoptotic cells[6,10,24].

Independent dimensionality reduction of each cell line with Phate captures the expected separation of the major G0G1, S, G2, and M CC phases defined by a classical gating strategy using IdU incorporation for S-phase, pH3 (Ser10), and CyclinB1 (Fig. 2C)[30]. We observe consistent patterns of CC molecules that capture CC progression in each cell line, such as higher PLK1 and pRb (S780) expression in S, G2, and M phases, a reduction of SLBP as cells exit from S through G2, and CDT1 licensing for DNA replication primarily restricted to G0G1 cells (Fig. 2C). Notably, the individual cell line CC embeddings were organized differently from each other—with NALM6 and U937 having more similar embedding structures, and 293T cells having the most different—along with the patterning and abundance of CC molecules suggesting differences in CC molecules across cell lines.

### Variability in CC molecules in phases is cell-type specific
Considering these differences in CC embeddings and the shifted molecular distributions across cell lines, we compared the mean expression of each cell line against the total mean of all cells for each molecular target from a random, balanced sampling of each cell line to study cell line variance (Fig. 2D). We observed a trend of HEL and

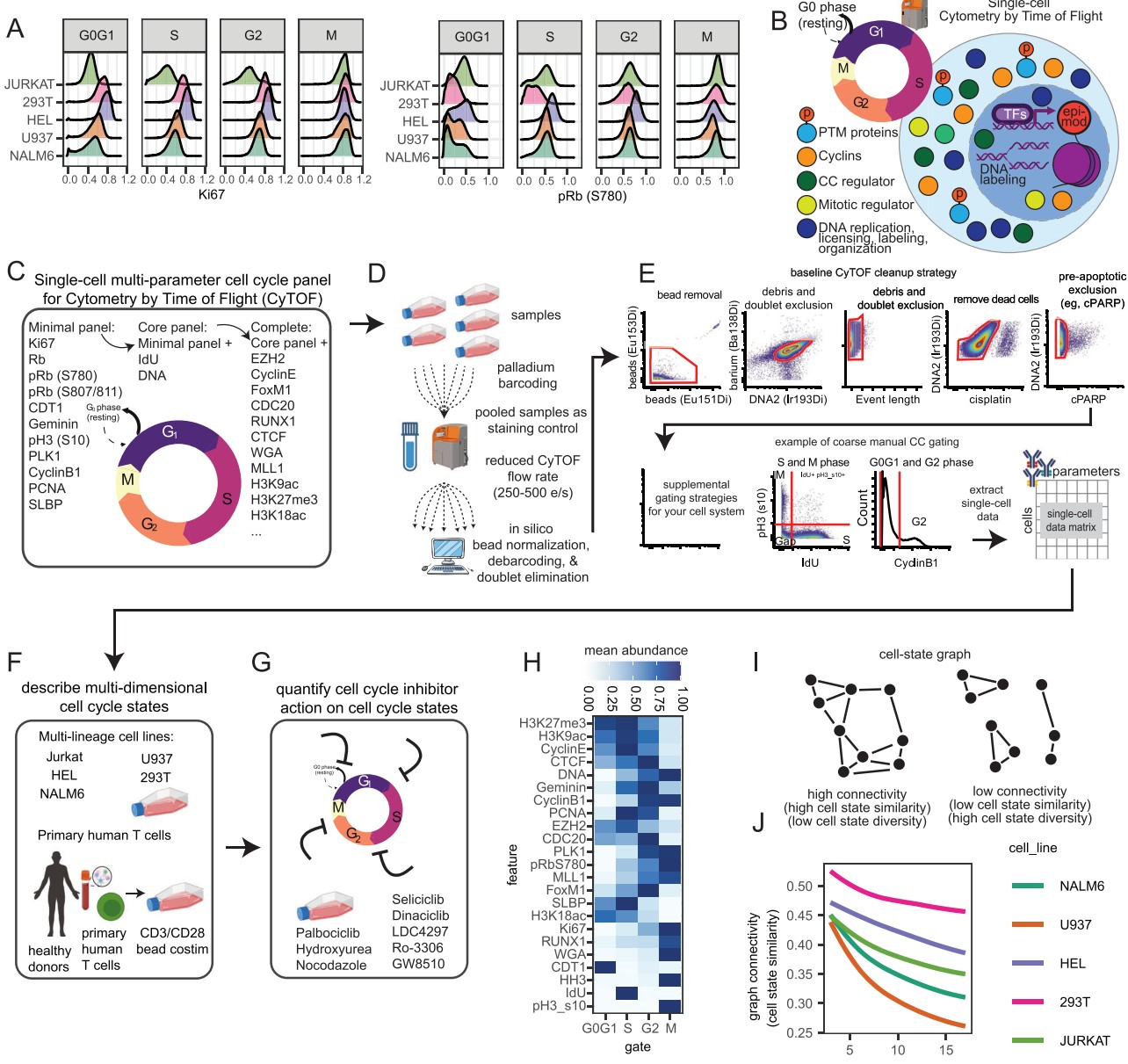

**Fig. 1 | A revised targeted approach to capture granular cell cycle information.**
**A** Example markers with CC distribution differences across cell lines facetted by coarse manually-defined groups. **B** Cartoon of different types of CC-related molecules. **C** Cartoon schematic for scMC panel of CC-related molecular targets for the minimal, core, and complete panels (see Supplementary Data 1 for all targets). **D** Cartoon schematic of controlled experimental multiplexing strategy for doublet reduction using palladium barcoding. **E** Recommended minimal gating strategy for data cleanup and example of coarse manual CC gating strategy. **F** scCC panel applied to cell lines with different cell-type identities and primary human T cells in proliferating and **G** CC perturbation settings. **H** Heatmap of CC-relevant markers used in cell line analysis (*n* = 3). **I** Schematic of the graph-based approach to quantify cell state similarity using edge density. Adjacency matrix computed using a cosine distance with a threshold of 0.5. **J** Mean connectivity (cell state diversity) as a function of # of features along all possible combinations of features in each cell line. Source data are provided as a Source data file.

293T cells having a globally higher abundance of CC-relevant molecules such as Ki67, PLK1, CyclinB1, and Geminin compared to NALM6, U937, and JURKAT cells. Notably, we observe the same patterns in WGA abundance and total histone content—proxies of cell size[24,31,32]. HEL and 293T cells tend to be larger (-12–15 μm) while the leukemic cell lines Jurkat and NALM6 cells are smaller (11.5 μm)[33,34]. However, U937—a pro-monocytic cell line—is larger (-13.29 μm) but does not express as much global CC molecule abundance as HEL and 293T[35]. Taken together, this suggests that while cell size may explain some of these differences unveiled by our scCC approach, the cell type origins and underlying origins of these cell lines likely explain the majority of variability in CC molecules across these systems.

With variation between cell lines evident, we were curious whether similar primary human cells from different individuals could exhibit similar traits. To investigate this, we queried primary human T cells from three different healthy individuals with our CC panel. Using primary human T cells sampled 96 h after ex vivo CD3/CD28 bead stimulation from three donors, we observed topologically similar CC embeddings between individuals, and similar to those seen for the cell lines (Fig. 2E, Supplementary Fig. 3A). CC phase occupancy defined by manual grouping averaged about 42% G0G1, 51% S, 5.1% G2 and 1.8% M, with little differences between donors (Fig. 2F). Considering there is known human variation in molecules across different cell systems, we did not observe

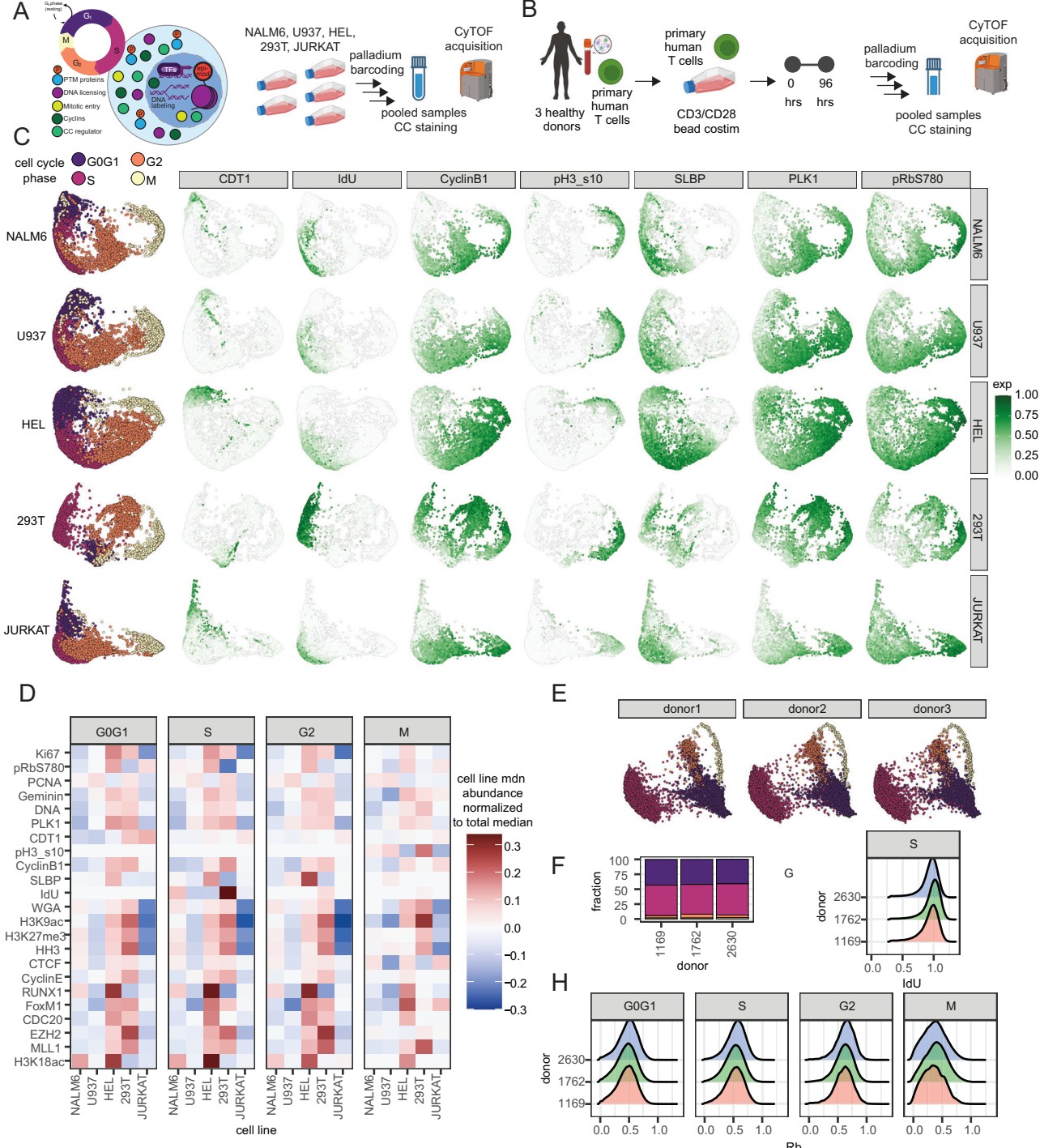

**Fig. 2 | CC embeddings and variance of CC information. A** Experimental design for CC analysis by CyTOF across palladium-barcoded cell lines (*n* = 3 each) pooled from multiple cultures. **B** Experimental design for ex vivo primary human T cell activation using CD3/CD28 dynabead stimulation across 3 unique donors. **C** Dimensionality reduction of cell lines and example markers from CC panel using PHATE. **D** Median molecular abundance of CC targets for each cell line, normalized to the median across all cell lines, each equally sampled. General CC phases are defined using CyclinB1, IdU, pH3(s10). **E** CC embedding molecular panel to CD3/CD28 dynabead stimulated primary human T cells from 3 donors. **F** Cell cycle phase fractions after 4 days of stimulation. **G** Donor variability in DNA replication measured by IdU incorporation during S-phase and **H** Rb expression across manually-grouped phases. Source data are provided as a Source data file.

differences in donor-dependent variation in CC-related molecule abundance like IdU, CyclinB1, pH3 (s10), and Rb. (Fig. 2G, H, Supplementary Fig. 3A). Our approach also captures expected abundance patterns, such as higher SLBP abundance in S, and a small subset of SLBP high cells in G2 as it undergoes rapid degradation upon G2 entry. PLK1, Ki67, and PCNA abundance increased with CC

progression, and pRb (S780) expression increased in both G0G1 and S-phase (Supplementary Fig. 3A). Taken together, our CC panel generalizes to diverse cell lines as well as primary human cells, with measurable differences in CC molecular abundance between cell lines despite the CC being a conserved molecular program intrinsically tied to a cell's biology.

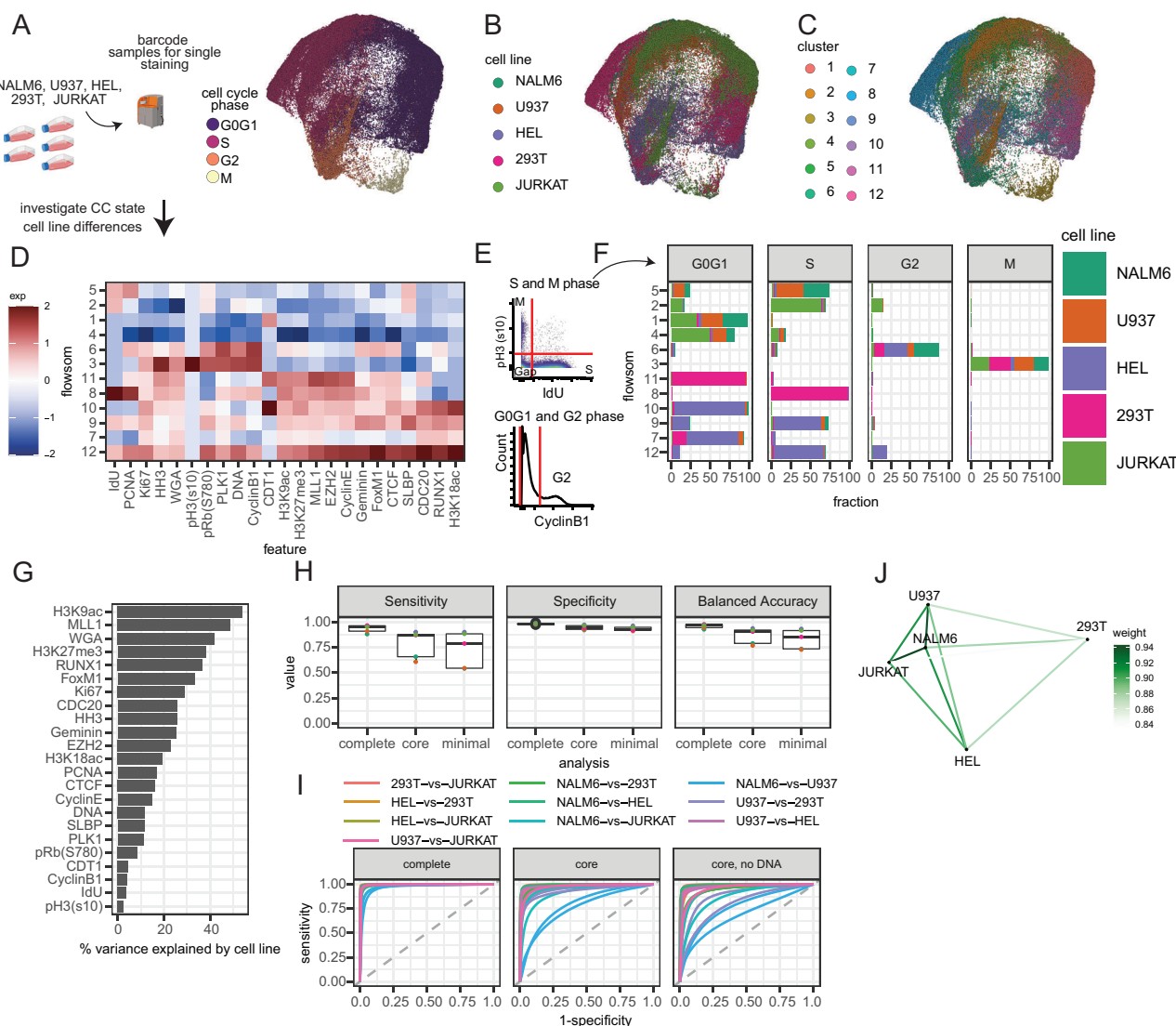

**Fig. 3 | Integrated multivariate CC analysis reveals differential cell-line CC states. A** CC embedding of molecular targets equally sampled for each cell line, pooled from multiple cultures, and z-scored across all cells in aggregate ($n = 4$ each across two experiments, showing 1 replicate each). Colored by (**A**) cell cycle group, **B** cell line, and **C** flowsom cluster. **D** Heatmap of CC molecular target abundance across clusters. **E** Current gold-standard gating strategy for identifying G0G1, G2, DNA replicating (S), and Mitotic (M) cells. **F** Fraction of cell lines in each cluster faceted by CC phase. **G** ANOVA results measuring % variance explained by cell line identity for CC molecular targets. **H** Sensitivity, Specificity, and Accuracy results for classifying single-cell cell line identity in a test set using multinomial logit regression with the complete, core, and minimal CC panels ($n = 5000$ cells each). Boxplots represent the median and the 25/75% interquartile range. **I** ROC curve using each panel. **J** Analysis of different CC molecular correlation patterns between each cell line, showing similarity between cell lines using the core panel. Source data are provided as a Source data file.

## CC quantification reveals distinct patterns across cell lines

Considering the variance of CC molecule abundance patterns we observed across cell lines was robust across replicates (Supplementary Fig. 2A, Supplementary Fig. 4A, B), we sought to leverage our deep CC phenotyping panel to further integrate and understand the molecular abundances in aggregate. Pairwise comparison of cell lines across all replicates identified NALM6 and JURKAT cells with the fewer CC-related molecular differences (Supplementary Fig. 4C). CC embeddings showed a coordinated separation of CC phases with clear cell line-dependent patterns (Fig. 3A, B). Single-cell data integration methods like Harmony are often used to correct batch or biological confounders, yet these approaches come at a tradeoff between batch removal and bio-conservation[36,37]. Employing Harmony to correct cell line differences adequately integrated cell line differences; however, CC phase purity analysis using FlowSOM for multivariate unsupervised clustering showed that while S-phase cells observed an improvement

in cluster purity after correction, G0G1 and G2 purity decreased (Supplementary Fig. 4C, D). Considering the tradeoff with using correction algorithms leading to the removal of biological signal or insertion of non-biological signal, and the lack of systematic single-cell comparison of CC states between diverse cell lines, we bypassed biological batch correction and further studied these cell line differences[38,39].

Deeper probing of unharmonized clustering shows that clusters 2, 8, 9, 10, 11, and 12 capture cell lines with CC phase specificity, except for clusters 9 and 12, which have mixtures of G0G1/S or G0G1/S/G2 phases. Clusters (1, 3, 4, 5, 6, 7) capture mixtures of cell lines with G0G1, S, and G2 phases defined by IdU, pH3(s10), and CyclinB1 expression. Interestingly, cluster 3 captured the majority of mitotic cells regardless of cell line status (Fig. 3C–F), which is consistent with previous studies looking at multivariate molecular measurements of chromatin state across CC phases in different cell lines[40]. Clusters (5, 6, 8, 9, 12) with

more proliferative phases (S/G2/M) had generally greater molecular abundance of key molecules like Ki67, PCNA, and Geminin, but cluster 2 is an interesting exception, capturing cells with active DNA replication (IdU+) cells that are Ki67 low. CDT1 positivity in clusters 1 and 10 indicates replication licensing before S-phase, but cluster 10 primarily consists of HEL cells with increased abundance of transcription factors like FoxM1 and CTCF, as well as SLBP, while cluster 1 has lower abundance of most CC-related molecules and includes diverse cell lines. High SLBP abundance in cluster 4 indicates deep progression into S-phase as SLBP coordinates histone synthesis with DNA replication, but again, this cluster is primarily detected in HEL and 293T cells. The majority of G2 cells are included in cluster 6, which has expected high expression of CyclinB1, PLK1, pRb (s780), and Ki67, but small fractions of G2 cells also appear in other clusters that are low in these molecules, typically expressed in G2 cells, such as cluster 2. SLBP loss is observed in clusters with larger G2 fractions (6,12), which is expected since SLBP degrades rapidly during the S/G2 transition[41]. In summary, cell line variance captured by the scMC CC platform is a convoluting factor in understanding CC states.

Considering cell line variance confounds CC with overclustering, we simplified the cluster task by reducing it to four clusters. There were mostly pure clusters of G2 and mitotic cells in clusters 3 and 4, while G1, S, and G2 phases were mixed between clusters 1 and 2 (Supplementary Fig. 4E). Whether under-clustering or overclustering, basal cell line differences in CC states complicate more granular CC analysis, and cell line variance is a strong contributor to CC state diversity. Integration may be necessary for experimental designs seeking to combine different systems, like cell lines. But the loss in signal associated with integration, the observed purity differences between CC phases, and the mixing of CC phase and cell line identities all demonstrate that our scCC approach captures disparate CC molecular programs that are distinct to each cell line.

## ML models quantify CC molecular abundance across cell lines

Considering the strong multivariate differences in CC molecular abundance across the cell lines, we sought to further quantify these molecular differences using interpretable univariate and multivariate statistical models that are balanced for each cell line in the training data. To estimate the contribution of each molecular target in separating the cell lines, we performed Analysis of Variance of simple linear regression models predicting molecular abundance from cell line identity (Fig. 3G). Molecules capturing chromatin state (H3K9ac, MLL1, H3K27me3) and cell size (WGA), as well as molecular regulators that are not necessarily CC-specific (RUNX1), had stronger associations explaining cell line differences (Fig. 3H)[24,31]. However, classical CC molecules like Ki67, CDC20, and PCNA also explained considerable variation. Consistent with the clustering results and literature analysis of chromatin state, pH3 (s10) poorly explained variance across cell lines, again suggesting that the phospho-chromatin state of mitotic cells is similar to each other[40].

To further quantify the differences in CC states in the cell lines, we trained a multinomial logistic regression model and evaluated its performance in predicting cell lines using the three CC feature sets from Fig. 1: (i) "minimal," (ii) "core," and (iii) "complete" (*molecules labeled in* Supplementary Data 1). Multivariate models predicting cell line outcome using all CC-relevant targets perform well with over 82% sensitivity, 97% specificity, and 96% accuracy across all cell lines, indicating robust cell line predictions (Fig. 3I, J). However, removal of chromatin state, cell size, and molecular regulators in the core panel reduces predictive performance, and the additional removal of DNA-related measurements in the minimal panel further reduces performance. Predictions using core CC markers achieve 75% accuracy, indicating that there remains fundamental differences in CC-specific markers across cell lines (Fig. 3I). Pairwise comparison of cell line predictions in the core and minimal panels further shows that

comparisons such as NALM6-vs-U937, U937-vs-293T, NALM6-JURKAT have poorer model performance, suggesting greater similarity between CC states due to less differences in CC molecular abundance between these cell lines (Fig. 3J). Correlation network analysis similarly suggests disparities in multivariate abundance patterns between cell lines, where JURKAT, NALM6, and U937 are most similar to each other. Notably, 293T—an adherent cell line—has the most different CC co-abundance pattern compared to the other suspension cell lines (Fig. 3I, J). Hierarchical clustering of CC abundance across replicates supports these results (Supplementary Fig. 4B).

We further investigated whether these CC differences are explained by single-nucleotide variant or copy-number variant profiles of CC-related genes using DepMap[42] published information for Jurkat, HEL, U937, and NALM6 (293T data were unavailable). Variant profiles were in agreement with the cell line heterogeneity observed in our scCC MC platform, though the majority of gene variants do not intersect with molecules probed in our platform (except PCNA) (Supplementary Fig. 4G–I, Supplementary Data 2 and 3). These results suggest that genetic sensitivity to CC-related variants may have gene-level biases based on cell line or evolutionary origins, which are detected indirectly by our scCC panel. While the CC is a generally conserved evolutionary program across different cell systems, we demonstrate that our approach can probe deeper into the diversity of cell states and reveal the subtle differences across similar but disparate CC molecular programs across diverse cell lines.

## CC analysis decouples canonical and noncanonical states

We demonstrated that deeper CC probing using our approach revealed the diversity of CC states across cell lines. CC states are often described by canonical rules like Ki67 expression as a marker for proliferative cells in S, G2, or M phase, and CDT1 licensing for DNA replication during G1. However, CC aberrancies are a hallmark of diseases like cancer, and these noncanonical CC states are reported across diverse, transformed cell lines[43]. Examples of noncanonical CC states induced by CC perturbation can include CDT1 overexpression in G2 leading to relicensing of DNA replication if unregulated by Geminin, failure to degrade SLBP during G2 leading to genotoxic stress, chromosome instability from various consequences such as loss of PLK1 during mitosis leading to mitotic slippage and cellular senescence, Ki67 loss or dephosphorylation of Rb while cycling, and other mechanisms that disrupt CC progression. Noncanonical CC states have been reported in cells experiencing DNA damage, mitotic infidelity, or disruption of CC regulators[19–21,28,44,45]. We observe that our scMC approach captures these noncanonical CC states without perturbation, such as low Ki67 abundance in S-phase cells actively replicating DNA, CDT1 expression during G2, high pRb (S780) or PLK1 expression during G0G1, low pRb (S780) in G2, and additional noncanonical cell states (Supplementary Fig. 2)[46]. To label noncanonical CC states, we discretized cells into canonical and noncanonical groups based on rules for canonical CC states found in the literature for features in our core panel and observed variable fractions of noncanonical CC states between cell lines and features (Supplementary Data 4). For example, canonical CC states with clear proliferative signatures are expected to express Ki67, or DNA licensing (CDT1) should not be abundant after S-phase, and thus deviancy from these cell states could indicate CC aberrancy. All cell lines had a subset of cells actively replicating DNA (IdU+) while Ki67 was low, CDT1-expressing cells in G2, and G2 cells that were pRb (S780) low, particularly in 293T cells (Supplementary Fig. 5A–C). Manually discretized noncanonical CC phenotypes were on average 23.9% of cells (NALM6, 14.6%; U937, 16.6%; HEL, 26.5%; 293T, 36.7%, and JURKAT, 25.3%) (Fig. 4A). Mahalanobis distance—a multivariate measure of each point (cell) from its population centroid, where larger magnitudes indicate further distance—of all cells in each phase was higher for noncanonical cells compared to canonical cells in the same phase, suggesting these grouped cells are distinct from the

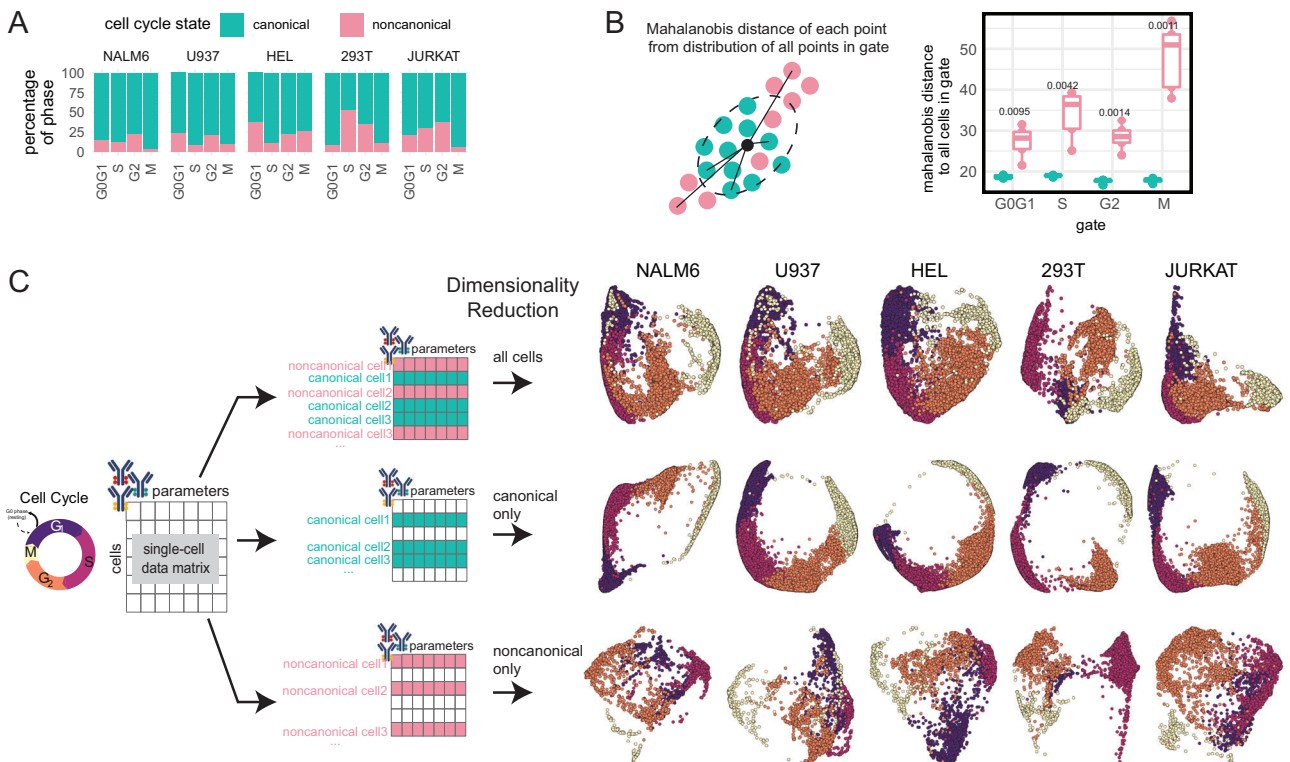

**Fig. 4 | Multi-parameter CC analysis captures canonical and noncanonical single-cell CC states in diverse cell lines. A** Estimated fraction of manually discretized noncanonical cells in each cell line and CC phase based on a manual discretization strategy. **B** Mahalanobis distance analysis comparing noncanonical cells and canonical cells to all cells ($n = 5$, two-sided Wilcoxon rank sum test).

Boxplots represent the median and the 25/75% interquartile range. **C** CC embedding was constructed on cells with both canonical and noncanonical, canonical-only, and noncanonical-only CC states using core panel features. Source data are provided as a Source data file.

average cell in each phase, with greater median aberrancy for noncanonical cells in S and M phase compared to their matched canonical counterparts (Fig. 4B). Considering that noncanonical CC states can be representative of perturbed CC progression or altered CC states, we asked whether including only canonical or noncanonical cells as input into dimensionality reduction would more accurately capture CC progression. Indeed, we observe improved topologically circular CC embeddings when excluding noncanonical cells, and aberrant topologies when excluding canonical cells (Fig. 4C). By more deeply measuring CC-related molecules at the single-cell level, we can reconstruct CC as historically learned through bulk assays as well as parse the diversity of CC states that may be over-generalized if studied through the lens of coarse CC phase annotation.

## CC perturbation induces noncanonical cell cycle states

While we quantified fractions of noncanonical CC states that reflect previously reported CC aberrancies, prior studies have also shown that pharmacologic inhibition disturbs the molecular state of CC progression[47–49]. Similarly, we used CC inhibitor assays to pharmacologically target different CC stages and study whether we can: (1) induce and capture known and unknown noncanonical scCC states; and (2) quantify inhibitor action. We treated Jurkat T cells with three different CC synchronization drugs for ~18 h and evaluated protein biosynthesis before harvesting cells for CyTOF using our platform after multiplexing samples with barcoding and removing dead and pre-apoptotic cells in silico (Fig. 5A, Supplementary Fig. 6A, Supplementary Data 5). We used Palbociclib (PALBO), a CDK4/6 inhibitor that primarily arrests cells in G0G1; Hydroxyurea (HU), an inhibitor of ribonucleotide reductase (RNR) that reduces the dNTP pool to slow down DNA polymerase movement at replication forks and activate the S-phase checkpoint; and Nocodazole (NOC), a microtubule disrupting

agent that binds to beta-tubulin and arrests cells at the G2M checkpoint, which can also induce mitotic slippage into G1[47–49].

Drug-treated cells piled up at expected CC phases, and CC embeddings suggested drug-treated cells occupy phenotypically distinct states from untreated (NO TX) cells (Fig. 5B, red circles). We further evaluated the inhibitory effects of PALBO, HU, and NOC at single-cell resolution by quantifying the perturbative action on CC relative to the untreated condition using MELD−a computational algorithm that uses graph signaling processing techniques to measure the sample-associated relative likelihood between experimental groups (Fig. 5D, Supplementary Fig. 6B)[50]. PALBO inhibitor action is primarily observed in G0G1 and a small subset of G2-like cells. HU inhibitor action is observed in G2, S, and G0G1 cells. NOC inhibitor action is observed in M and G0G1 cells (Fig. 5D). Inhibitor action on CC states is largely unique to each inhibitor, though some cells with G0G1 CC states induced by PALBO are also observed in NOC-treated cells, and vice versa. Scoring CC phase enrichment defined by manual groups further quantified inhibition action in treatment conditions, and our platform clearly captured noncanonical CC states induced by drug treatment in arrested cells, as well as cells that escaped arrest (Supplementary Fig. 6C).

We previously quantified noncanonical CC states using manual discretization, but this relies on coarse gating practices. To more deeply characterize induced noncanonical CC states relative to an unperturbed system, we used nearest neighbor (NN) analysis to quantify the multivariate Euclidean distance of each cell to its nearest untreated Jurkat T cell neighbors according to their CC molecular profile (Fig. 5E). NN analysis captures noncanonical cells with skewed phenotypic distance in treatment conditions compared to WT cells (Fig. 5F). Using regression analysis, we model the euclidean aberrancy score to identify aberrantly expressed features from each treatment.

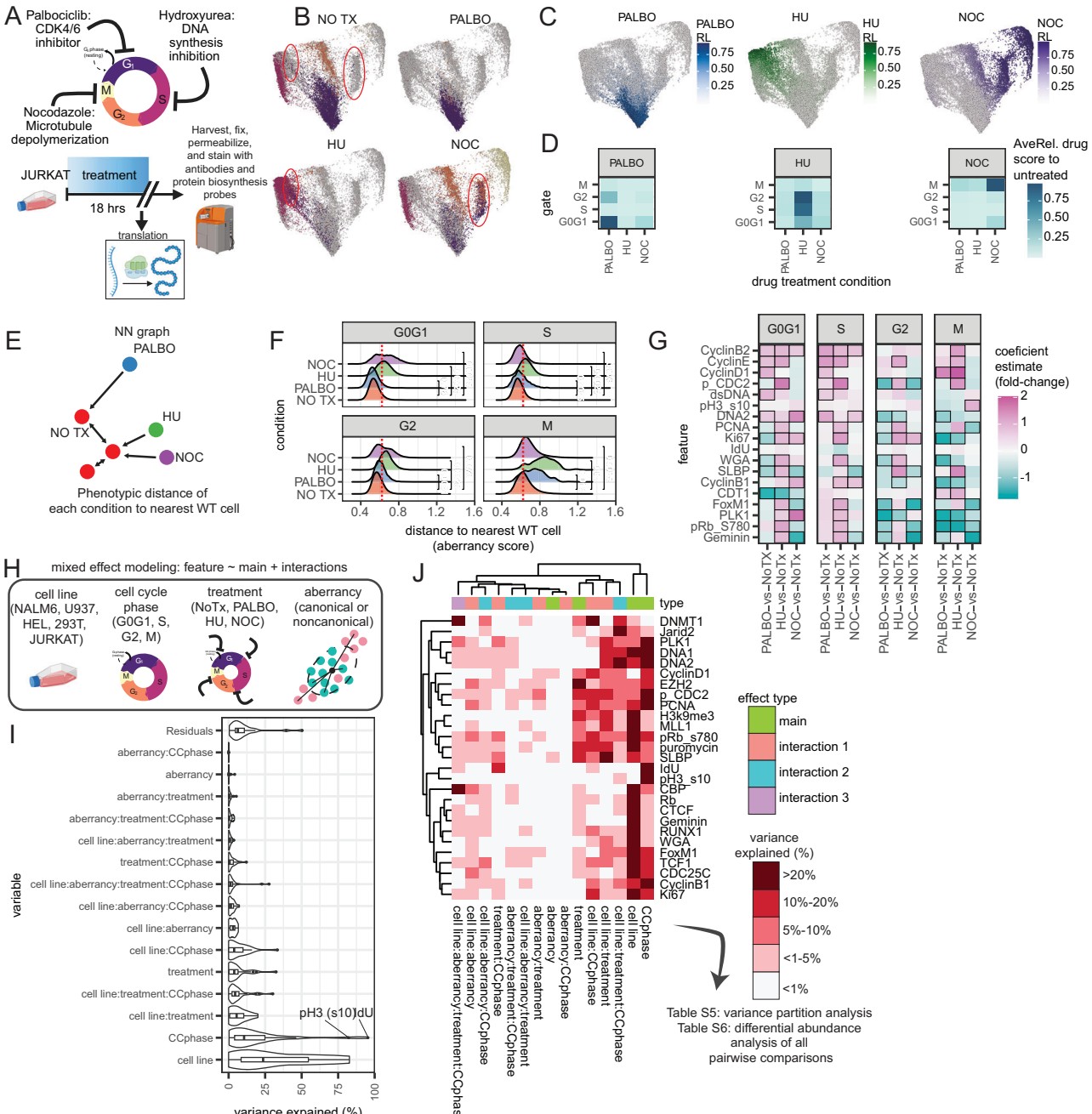

**Fig. 5 | CC inhibitors induce noncanonical CC states. A** Cartoon schematic of pharmacologic inhibition in JURKAT cells (n = 5 from 3 experiments). **B** CC embeddings for each drug with all cells in a gray background, highlighting induced CC states (1 representative sample). **C** CC embeddings with perturbation score quantified using MELD. **D** Median perturbation score for each group in each drug condition. **E** NN analysis cartoon schematic quantifying phenotypic CC distance from untreated cells. **F** Distribution of CC distance, dotted line indicates a manually-defined threshold for canonical cell identity based on the WT distribution across. Two-sided Wilcoxon test on mean aberrancy from each sample. **G** Heatmap results comparing drug treatment effects to NoTx (black box indicates significance

defined by padj ≤ 0.1 and absolute coefficient estimate ≥0.5 using generalized linear models and the Wald test. **H** Cartoon schematic for single-cell variance partitioning analysis across cell lines, cell cycle phase, drug treatment, and aberrancy (n = 36 unique samples, 3,544,481 single-cell measurements: 3 replicates each cell line, 3 replicates of each drug treatment in Jurkat cells, 1 replicate of each drug treatment in NALM6, U937, HEL, 293T cells). **I** Variance explained for model variables from the partition analysis using the pseudobulk samples (n = 36). Boxplots represent the median and the 25/75% interquartile range. **J** Binned percentage of variance explained by each main or interaction effect for each molecular target. Source data are provided as a Source data file.

Discovered features that represent noncanonical CC states include increased DNA content reflecting mitotic slippage of G0G1-phase NOC-treated cells; decreased pRb (S780) in G2-phase PALBO- and NOC-treated cells but M-phase PALBO- and HU-treated cells; Ki67 loss in S-phase PALBO- and NOC-treated cells, indicating proliferative aberrancy; and increased PLK1 expression in G0G1-phase NOC-treated

cells, and a loss of PLK1 in G2-phase PALBO-treated cells (Fig. 5G). These noncanonical skewed cells are clearly observed in lower-dimensional CC embeddings of each CC phase (Supplementary Fig. 6F). Considering these are cell lines with demonstrated non-canonical CC states, canonical cells defined using the NN distance threshold that still demonstrate skewed phenotypes from the

untreated setting exist because of short Euclidean distances to natural-occurring noncanonical cell states in the untreated cells.

Finally, we integrate protein biosynthesis analysis using puromycin incorporation and detection with our scCC platform to further understand whether noncanonical cells induced by drug treatment affect protein biosynthesis. Consistent with what we have previously reported, we found de novo protein synthesis active in G0G1 through G2/M[8]. Here, we observe that protein translation is reduced because of CC inhibitor treatments, particularly in NOC-treated cells (Supplementary Fig. 6G), with affected cells enriched in all stages except S-phase. This loss in protein translation is skewed among noncanonical CC states induced by treatment that are phenotypically more distant from WT cells (Supplementary Fig. 6H). Taken together, our scCC approach not only captures the landscape of CC states that arise from pharmacologic inhibition but also flexibly integrates with other scMC technologies like de novo protein synthesis detection to reveal the link between protein translation and CC inhibitor action.

### Multi-systems variance partitioning using mixed modeling

Considering our platform detects single-cell heterogeneity across cell line, CC phase, CC inhibitors, and CC aberrancy, we generated a massively parallel single-cell dataset of multiple cell line replicates and drugs in each cell line to quantify the main and interaction effects across cell line identity, drug treatments, CC phase, and CC aberrancy using mixed effects modeling (Fig. 5H, Supplementary Fig. 7A–C). Variance partitioning revealed that cell line identity drives the most differences, followed by CC phase, and moderate contribution due to CC aberrancy or treatment effects that interact with cell line identity (Fig. 5I). For all CC measurements, we binned variance explained in each coefficient into <5%, 5–10%, 10–20%, and >20%. Notably, many transcription factors involved in CC regulation, like RUNX1, EZH2, TCF1, FoxM1, and Jarid2, were partially explained by CC aberrancy (Fig. 5J, Supplementary Fig. 7B). Cell-line-dependent drug sensitivity was detected in DNMT1, FoxM1, WGA, EZH2, pRb, SLBP, pCDC2, and DNA content. Molecules like SLBP, PCNA, pCDC2, pRb, MLL1, and de novo protein synthesis also demonstrated cell-line-dependent variance with treatment or phase. PLK1, DNMT1, and EZH2 had partially explained variance by interactions across cell line, phase, and aberrancy. This analysis demonstrates the complexity of how these different biological factors are layered to create heterogeneity in CC states. IdU and pH3(s10) were used to manually discretize S and M phase cells and had high variance for CC gates, which may mask their contribution to other variables. Notably, many transcription factors involved in CC regulation, like RUNX1, EZH2, TCF1, FoxM1, and Jarid2, were partially explained by CC aberrancy (Fig. 5J, Supplementary Fig. 7D, E). Molecules like SLBP, PCNA, phospho-CDC2, and de novo protein synthesis also demonstrated cell-line-dependent variance with treatment and phase (Supplementary Fig. 7B). Variance partitioning and differential abundance results are shared in Supplementary Data 6–8. In summary, our platform enabled us to quantify the effects of different biological factors involved in scCC state diversity.

### CC inhibitor in ex vivo stimulated primary human T cells.

Although the induction of aberrant cells with noncanonical CC states using CC inhibitors is an observed phenomenon with adherent cells using microscopy, transformed cell systems like tumor cell lines have documented aberrant CC behaviors compared to healthy primary cells. Thus, the induction of aberrant cells with noncanonical CC states using CC inhibitors may be exacerbated by CC perturbation[45]. Healthy primary human T cells from the peripheral blood are the majority quiescent suspension cells that undergo rapid expansion upon activation and have been a recent focus of ex vivo expansion efforts for their various clinical applications, emphasizing the importance of deeper CC analysis to understand the therapeutic relevance of expanded cell products[51]. Thus, deeper CC phenotyping may help us

understand how different drugs affect the same system as well as the canonical and noncanonical CC behaviors that arise in healthy T cell settings. To better characterize the effects of CC inhibition on normal and aberrant CC states in healthy cell systems, we use ex vivo TCR stimulation of primary human T cells treated with various CC inhibitors from Days 0 to 3 of activation (Fig. 6A). All drugs achieve CC arrest after 3 days of stimulation that more closely resembles the CC state of D0 unstimulated T cells than the D3 cells, except for Ro-3306, which still demonstrates inhibitor action by a reduction in the abundance of DNA licensing factor CDT1 (Fig. 6B, C).

### T cells escaping CC inhibition have more aberrant CC states.

To further understand the effects of these drugs on scCC states in healthy human settings, we quantify CC state aberrancy and inhibitor action using NN analysis. Similar to cell lines, we observe an enrichment of noncanonical CC states as a consequence of pharmacologic inhibition of CC (Supplementary Fig. 8A). For example, Ki67 loss in G2 cells, a reduction in DNA replication measured by IdU incorporation in LDC4297 and Palbociclib, a reduction in PCNA for Dinaciclib and Palbociclib, and PLK1 reduction in M cells across all treatments. Drug perturbation scores normalized to the untreated D3 cells shows Palbociclib, Seliciclib, Dinaciclib, and LDC4297 with strong arrest signatures at G0like and G0G1 CC states (Fig. 6C–F, Supplementary Fig. 8B). Correlative analysis of raw scores to compare single-cell inhibitor action between the different treatments shows that Palbociclib, Dinaciclib, LDC4297, and Day 0 NoTx CC states are strongly correlated with each other, and Seliciclib as distinct from other treatments (Fig. 6G). Despite the strong correlations between Palbociclib, Dinaciclib, and LDC4297, these correlations are not perfect, and there remains some CC state diversity that is unique to each drug. While GW8510 and Ro-3306 did have a minor effect on CC states, the drug CC states were still strongly correlated with D3 NoTx cells. Notably, normalizing inhibitor action for each cell to the D3 NoTx score shows that the cells affected with shared signatures with each drug were strongly correlated, with some unique CC state diversity as a result of each inhibitor (Supplementary Fig. 8C–F). To better understand this CC state diversity, we further analyze the aberrancy of cells that escape inhibitor action by quantifying the phenotypic distance of CC states for all cells to any untreated cell at Day 0 or Day 3 (Fig. 6H). Aberrancy analysis revealed that cells escaping inhibitor action have CC states that are disparate from untreated cells, and are more evident in cells that have S and G2 signatures (Fig. 6I, Supplementary Fig. 8G). Taken together, this analysis suggests that while CDK inhibitors do induce some unique diversity in CC states during ex vivo TCR stimulation of healthy primary human T cells, they have globally similar inhibitor action, and that cells escaping arrest demonstrate more aberrant CC states.

In summary, our scCC approach reveals the utility of combining deep scCC phenotyping with pharmacologic inhibition to guide our understanding of how different drugs affect a system. By looking specifically within quiescent healthy human primary cells that proliferate only upon stimulation, we can study the canonical and noncanonical behaviors induced by CC inhibition in a system where cell proliferation and expansion are relevant for producing therapeutic cell products. Our results also suggest that pharmacologic control of CC may be used to tune proliferation during the stimulation phase of manufacturing therapeutic cell products.

## Discussion

The effects of CC state and proliferation pervade diverse systems like cell differentiation, cellular function, and cancer. As technological advances in single-cell proteomic measurement capabilities expand and enable multi-systems analysis, there is a need to increase measurements of CC biology to more deeply characterize CC directly, along with the crosstalk between CC and other systems. We greatly

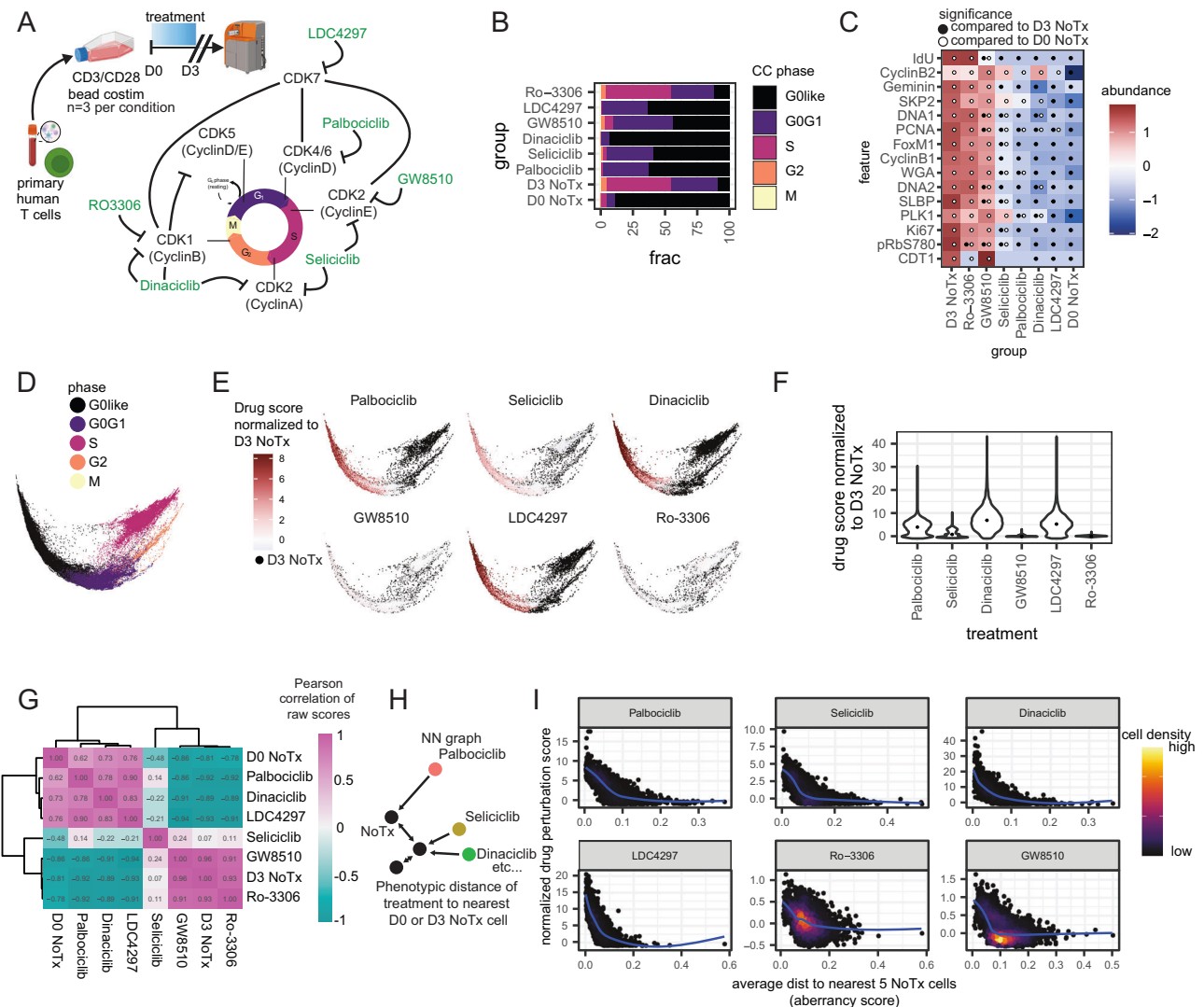

**Fig. 6 | CC aberrancies are associated with escape from CC inhibitor action on ex vivo stimulated primary human T cells. A** Cartoon schematic of experimental design using ex vivo stimulated primary human T cells from a healthy donor with various selective CDK inhibitors ($n$ = 3 per condition). **B** Example CC phase fraction for each treatment. **C** Median abundance of CC features. Significance defined by padj ≤ 0.1 and absolute coefficient estimate ≥0.5 (features were mean-centered and scaled). Black dots indicate significance compared to D3 NoTx, and white dots

indicate significance compared to D0 NoTx. **D** CC embedding of all cells colored by group and **E** colored by drug score, normalized to NoTx (MELD) for each treatment. **F** Relative drug scores for each treatment normalized to D3 NoTx. **G** Pearson correlation coefficient between raw drug treatment scores using pseudobulk values across replicates (all padj ≤ 0.00001, except Seliciclib = 1). **H** Cartoon schematic of NN analysis. **I** Drug perturbation score and CC aberrancy score for each treatment, colored by cell density. Source data are provided as a Source data file.

extended existing approaches for cytometric CC analysis by capturing the expression of new proliferation molecular regulators, licensing factors, CC checkpoint inhibitors, and chromatin states. By combining this molecular panel with barcoding strategies to increase throughput and control for technical effects, we demonstrate the large-scale capture and quantification of diverse CC states across millions of cells and multiple cell lines with pharmacological perturbation. We further demonstrated the generalizability of our molecular panel in CC-arrested primary human T cells.

While CC progression is controlled by the synthesis and degradation of cyclins in an oscillatory manner, CC checkpoints serve to regulate this molecular orchestra. Our understanding of canonical CC progression has been historically shaped by bulk assays like western blots or low-throughput, low-dimensional imaging assays[12,52]. These canonical definitions inform normal CC progression but do not fully capture the single-cell landscape of CC states. With the advent of sc technologies, CC remains coarsely studied aS-phases, CC signatures are often regressed out instead of directly studied, and is limited by

adherent systems from imaging platforms, or CC biology is inadequately captured because it is fundamentally a protein- and PTM-mediated process, leaving much to be desired for deep CC phenotyping in sc assays like scRNA-seq[53,54]. However, aberrant CC states are a reported phenomenon in perturbation systems—microtubule destabilization and cancer are known to induce noncanonical CC states[49,55]. Using our CC molecular panel, we capture the diverse single-cell CC states naturally abundant in cell lines that do not qualify canonical, discretized CC states defined in the literature, as well as pharmacologically-induced noncanonical CC states. For example, Ki67—often used to identify proliferative cells—was a dynamic molecule with low expression in a subset of Jurkat cells with other molecular signatures of proliferation, but also decreased across all phases in the presence of CDK4/6 inhibition. CyclinB2, but not CyclinB1, increased in G0G1 cells with all treatments, which is particularly interesting because both activate CDK1, though CyclinB2 is thought to have a less important role for viability, but can compensate for CyclinB1[56]. SLBP, typically expressed in S-phase, was increased in G0G1 and G2 cells treated

with HU. We've demonstrated that our scCC platform captures these nuanced cell abundances, is amenable to large-scale high-throughput suspension and adherent cells, and can be easily integrated with other cell profiling approaches, all of which can be easily extended to dissociated tissues.

While we observed noncanonical cells naturally existing in cell culture systems, cell lines are transformed with atypical proliferative behaviors, if not already derived from cancer cells[57–59]. Thus, it is possible that the induction of noncanonical CC states is an artifact of transformed systems known to have CC aberrancies. To answer this, we further assessed the effectiveness of diverse CDK inhibitors that target a range of CDK molecules controlling different CC checkpoints in CD3/CD28 costimulated primary human T cells from healthy donors. While CDK inhibitors had quantifiable differences in the induced scCC states, the effects of CDK inhibition primarily resulted in G0G1 arrest, and drug action was strongly correlated for Palbociclib, Dinaciclib, and LDC4297. Notably, primary human T cells that escaped inhibitor action had more phenotypically aberrant CC states when compared to untreated T cells, demonstrating that noncanonical CC states are pharmacologically inducible in healthy systems as well. Our understanding of noncanonical CC states will continue to evolve as we more deeply understand the dynamics and roles of CC molecules in healthy and perturbation settings.

In summary, our CC-related molecular panel can capture the diversity of canonical and noncanonical scCC biology in both cell line and primary cell systems, which can be further integrated with perturbation systems to dissect granular CC biology. This is consistent with previous single-cell transcriptomic studies showing that CC drug-tolerant PC9 cells occupy heterogeneous and distinct cell subpopulations and cell states[60]. These distinct cells that escape early CC arrest and survive may also be noncanonical seed cells for drug-tolerant cells. In the case of primary cells, cell proliferation is an essential metric for therapeutic cell products. Previous studies demonstrate that cell proliferation and cell fate are tightly linked in T cells, where tuning TCR stimulation has a direct impact on CC progression and division[26]. Extending our scCC platform to T cells, we demonstrated that CC slowing with CDK inhibition has direct consequences on CC state, which future studies can harness to understand whether CC tuning is useful for the manufacturing process of cell therapeutics.

We present a robust and generalizable platform for parallel scCC measurements using CyTOF to deeply characterize CC states that future studies can leverage in tandem with other MC molecular panels for sc and dissociated tissue systems. Multiplexing our scCC panel with published or custom panels such as epigenetic state using EpiTOF, chromatin content using Chromotyping, single-cell metabolic regulome profiling (scMEP), immune cell monitoring, or cell differentiation in perturbation and disease settings can be valuable to dissect the crosstalk between CC and other cellular processes in different areas of life science research.

## Methods

### Ethics statement

All research complies with relevant ethical regulations. De-identified peripheral blood and LRS chambers samples from healthy human donors were purchased and obtained as de-identified samples from Stanford Blood Center, and experiments were carried out following the guidelines of the Stanford Institutional Review Board (IRB). Collections were monitored and reviewed by Stanford's IRB. Written informed consent for collection and research use was obtained from all participants managed by Stanford Blood Center. Age, sex, and/or gender of donors were not acquired from samples after de-identification because this study is a development of technologies where age, sex, and/or gender are not variables to assess. Additionally, sex and gender information is inaccessible for donor samples

purchased from Stanford Blood Center. Genomics variant data were collected from published data resources, and all data reported were freely accessible from DebMap.

### Cell line culture

Five cell lines were used in this study: NALM6 (ATCC CRL-3273), U937 (ATCC CRL-1593.2), HEL (ATCC TIB-180), 293T (ATCC CRL-3216), JURKAT (ATCC CRL-2899). NALM6 is a lymphocyte-like cell line derived from human B cell precursor leukemia, U937 is a promonocytic cell line originating from human myeloid leukemia, HEL is an erythroblast cell line derived from human erythroleukemia, 293T is an adherent epithelial-like cell line originating from human embryonic kidney cells, and JURKAT is a CD4+ lymphocyte-like cell line derived from human T cell acute leukemia. The NALM6, JURKAT, U937, and HEL cells were maintained in RPMI 1640 (Gibco, 11-879-020), 10% FBS (Sigma-Aldrich, F4135), 1% p/s (Thermo Fisher Scientific, 15140122), and Glutamax (Thermo Fisher Scientific, 35-050-061). 293T cells were maintained in DMEM (Dulbecco's Modified Eagle's Medium)/Ham's F-12 50/50 Mix (Corning, 10-090-CV), 10% FBS, and 1% p/s. Cells were maintained at 37 °C, 5% $CO_2$.

### Ex vivo primary human T cell TCR stimulation

Please see the ethics statement for human sample collections. PBMCs were isolated via Ficoll (GE Healthcare) density gradient centrifugation. Bulk T cells were negatively isolated from whole blood using RosetteSep Human T Cell Enrichment Cocktail (StemCell Technologies). T cells were cultured in ImmunoCult-XF T Cell Expansion Medium (10981, Stemcell Technologies) and supplemented with 10 ng ml$^{-1}$ of interleukin-2 (Miltenyi Biotec). T cells were activated and expanded using Human T-Expander CD3/CD28 (Dynabeads, Thermo Fisher) added in a 1:1 cell-to-bead ratio, and cells were incubated at 37 °C in 5% $CO_2$.

### IdU labeling

5-Iodo-2-deoxyuridine (Sigma I7125) was re-suspended in DMSO (Sigma D2650) at 500 mM. IdU labeling was performed at a final concentration of 100 μM and returned to the incubator for 15–30 min. After labeling, cells were washed with PBS and continued with the live-dead labeling and fixation protocol for CyTOF.

### Live/dead labeling, cell fixation, and permeabilization for CyTOF staining

CyTOF staining was performed following this protocol.io publication[61]. Briefly, live-dead labeling was performed using cisplatin in low-barium PBS. Cells were washed with CSM, fixed with PFA for 10 min at room temperature, and stored in −80 °C until staining. When ready, fixed samples were thawed at room temperature and barcoded using fixed-cell palladium barcoding, pooled into a single tube. Surface staining was performed with metal-conjugated antibodies in CSM for 30 min at room temperature, washed, permeabilized with 100% 4C methanol, and incubated on ice for 10 min, washed 3× times with CSM, and proceeded with intracellular staining. Finally, cells were washed with CSM and re-suspended in an intercalator solution until CyTOF analysis.

### Drug treatments

Drug treatments were performed using ibrutinib (PCI-32765; Cellagen Technology #C7327), palbociclib (Chemscene, CS-3110), roscovitine (Seliciclib, CYC202, Selleck Chemicals #S1153), dinaciclib (SCH727965, Selleck Chemicals, #S2768), LDC4297 (LDC044297, Selleck Chemicals, #S7992), ro3306 (Selleck Chemicals,#7747), GW6510 (sc-biotech #sc-215122), hydroxyurea (Sigma-Aldrich #H8627-1G), nocodazole (Sigma-Aldrich. #SML1665-1ML), SU9516 (EMD Millipore, Calbiochem, #572650). Concentrations are provided in Supplementary Data 5.

## Palladium barcoding and staining with metal-conjugated antibodies

Individual samples within one experiment were palladium barcoded as described previously[14] and combined into a single sample before further processing and staining. Experiments with multiple barcode plates had an anchor sample to normalize any technical effects.

## CyTOF processing

Raw MC data were bead-normalized to remove acquisition-related influences on marker expression using the premessa R package. Sample barcoding was done using fixed cell palladium barcode combinations and debarcoded using Premessa. Normalized data were uploaded to CellEngine for bead removal, singlet identification, removal of debris and non-biological events, live cell gating, and exported (https://www.cellengine.com/). Pre-apoptotic cells defined by cPARP positivity were also removed. Batch correction between multiple palladium barcodes in the same experiment was performed using an anchor sample with an adjustment factor for each channel[62]. Adjusted data were imported into CellEngine for manual CC gating using CyclinB1, IdU labeling, and phospho-Histone H3 (s10). Gated files were subsequently imported into the R environment, asinh transformed (cofactor 5), and normalized to the 99.9th percentile of each respective channel before downstream analysis.

## Dimensionality reduction, quantifying perturbations, and diversity

We used PHATE (R/Python) with knn_dist=cosine, mds_dist=euclidean, and knn=15. We used RANN::nn2 in R or sklearn's KNeighborsClassifier in Python to find the nearest WT or untreated cell neighbors by Euclidean or cosine distance. Single-cell perturbation scores were quantified using MELD with default parameters. Analysis and plotting were done in R v4.2. For cell state diversity, we downsampled to 2000 cells per cell line and used Ki67, pRbS780, pH3_s10, CDT1, IdU, Geminin, PLK1, DNA, CyclinB1, PCNA, SLBP, CyclinE, WGA, HH3, EZH2, CTCF, and MLL1. We computed an adjacency matrix for all possible combinations of features using a cosine distance with a threshold of 0.5 and quantified graph density using igraph. Statistical models were computed using a random, balanced sampling of 50,000 cells per cell line. Package versions and features used are described in Supplementary Data 9 and 10.

## Statistical analysis

We use a generalized linear model/mixed model (GLM/M) and Wald test framework as used with diffCyt for statistical comparisons unless otherwise noted in the figure legends. All tests are two-sided. Multiple hypothesis correction was performed in R using p.adjust() with a False Discovery Rate (Benjamini-Hochberg) correction procedure. Features were mean-centered and scaled prior to statistical analysis. Proportion of variance calculations for cell line associations were performed using lm and anova in R. The sum of squares was calculated, and the proportion of variance explained was computed from the total sum of squares. Multivariate logit models were computed using nnet::multinom in R, and pairwise sensitivity, specificity, and accuracy were calculated using pROC::multiclass.roc. Variance partitioning was performed using the variancePartition R package and including main effects as well as all two-way, three-way, and four-way interaction effects across cell lines, drug treatment, aberrancy, and CC phase. Detailed code and downsampled example data for linear modeling are provided in: https://github.com/mamouzgar/2025_cellcycle_ml

## Visualization

Data figures were plotted using ggplot2 v3.5.2. Cartoon schematics were generated using a combination of BioRender and Adobe Illustrator. A license for BioRender figures is supplied: Created in BioRender. Amouzgar, M. (2025) https://BioRender.com/lfwmxok

## Reporting summary

Further information on research design is available in the Nature Portfolio Reporting Summary linked to this article.

## Data availability

All single-cell data (FCS files) generated in this study have been deposited in the Zenodo database under accession code: 10.5281/zenodo.14852934 [https://zenodo.org/records/14852934] without any restrictions. Analysis files are provided as Supplementary Data files. Cell line variant information was gathered from the published DepMap database. The analyzed SNV and CNV data extracted from DebMap are provided as Supplementary Data Files. Source data are provided with this paper at version 2 of the same Zenodo database: 10.5281/zenodo.14852933 [https://doi.org/10.5281/zenodo.14852933]. Any additional information desired in this article is available from the lead contact upon request.

## Code availability

Example code for analysis is available at https://github.com/mamouzgar/2025_cellcycle_ml/tree/main.

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

## Acknowledgements

Stanford Bio-X SIGF, Felix and Heather Baker Interdisciplinary Graduate Fellowship, Stanford Immunology T32 Training grant #T32 AI007290-37, and Stanford Biosciences Scholar Fellowship: M.A. This work was supported by NIH grants DP2EB024246, U24CA224309, R01AG068279, U54HL165445, R01AG078702, R01AG088656, P01AG036695, R01AI189963: S.C.B. The authors thank Jolene Ranek for manuscript comments and suggestions.

## Author contributions

Conceptualization: M.A. and S.C.B. Methodology: M.A. Software: M.A. Computational analysis: M.A. Experimental data curation: M.A. Published data curation: M.A. Mass-tagged antibody conjugations: T.B. Blood orders and T cell isolation: D.H. and P.F. Writing: M.A. and S.C.B. Editing: M.A., S.C.B., and P.F.

## Competing interests

The authors declare no competing interests.
