## [Transparent Peer Review file · Nature Communications]

A deep single cell mass cytometry approach to capture canonical and noncanonical cell cycle states

Corresponding Author: Professor Sean Bendall

Version 0:

Reviewer comments:

Reviewer #1

(Remarks to the Author)

In the manuscript by Amouzgar et al., the authors have developed a very detailed CyTOF panel for cell cycle analysis. This is a great addition to the field, and interestingly the authors demonstrate that it can be used to classify cells into canonical and noncanonical cell states. The manuscript is well-written, and the figures are generally well-structured. The analysis takes a broad, global approach by examining all markers simultaneously, which provides a holistic overview. However, currently it is difficult to tell which specific cell cycle-related markers are most informative, particularly in relation to cell line heterogeneity, drug sensitivity, and distinctions between canonical and noncanonical cell cycle states. I would thus recommend that the authors refine the analysis to highlight key marker differences by using additional statistical comparisons. Additionally, to maximize the impact of this methods paper for the community, a detailed R analysis pipeline should be provided, specifically outlining the steps for distinguishing canonical from noncanonical cell cycle cells. Furthermore, the manuscript appears to lack statistical comparisons and validation of technical reproducibility (e.g., technical replicates for cell lines). While I do not suggest additional experiments, further analysis of the existing data and improved documentation would strengthen the study. Here are my additional comments:

- The authors attempt to explain differences in cell cycle-relevant molecules based on cell size, however, they do not identify a specific trend. It would be worthwhile to investigate which mutations in cell cycle-related molecules are present in these different cell lines using publicly available data. Does the mutation status help to explain the observed heterogeneity among cell lines?
- It would be great to have an overview cartoon figure for the function of each marker in the panel.
- The differences in cell cycle markers between cell lines are very interesting, and I agree with the authors' decision not to use Harmony to correct for these differences. However, I think this should rather be a technical note in the methods, as it is taking attention away from the main story line. But, I let the authors decide what they think is better.
- Statistics and number of replicates is missing from the figure legends.
- One of the advantages of this technique is that we can find new and interesting differences between cell lines. But to do so, we need to consider technical variation of the cell lines. From the fcs file name, it seems that three replicates were pooled into one file. I would like to see an assessment of how the cell lines vary individually. Were the cell lines synchronized in each experiment?
- Likewise, what is the technical variability of the drug treatment, i.e. how does the replicates of the experiments look like. Comparisons should also be done on a statistical basis with e.g. diffcyt or findmarkers (from scan).
- It would be optimal if the authors could share their R analysis pipeline for the CyTOF data, but if that is not desirable, then at least we would need information on which markers were used for generating the UMAPs and FlowSOM clusters.
- The comparison of cell lines in Fig. 3 is too holistic to understand what the main differences are. While all markers are necessary to fully distinguish the cell lines, it remains unclear which differences are most critical. Which markers are predominant in each cell line, and why? To address the "why", I suggest examining the mutational status, as previously recommended.
- The division of cells into canonical and noncanonical states is very interesting. Could the authors try trajectory analysis (e.g. PAGA tree) or similar on the cell cycle markers to estimate at which (canonical) cell cycle stage the noncanonical cell cycle stages are generated? It would also be important to describe how exactly the cells are divided into canonical or noncanonical using example R code.
- "Here we greatly extended existing approaches for cytometric CC analysis by capturing the expression of new proliferation molecular regulators, licensing factors, CC checkpoint inhibitors, and chromatin states". There are indeed a lot of interesting markers in the panel, but there is almost no mention of any of them in the results or the discussion. Often when new CyTOF

panels are adapted in other labs, we do not use the entire panel, but the key extra markers. So, which are the most important markers that could be used in distinguishing cell heterogeneity, drug sensitivity, and non-canonical from canonical cells?

Minor corrections

- Fig. 1D, Fig. 3A, Fig. 5A, Fig. 6G text is too small.
- What is 1 degree T cells in Fig. 1E?
- "cell line lineages" should be "cell lines".
- Version numbers for packages used in R are missing from the methods.
- The R script provided runs, but it could be better annotated for non-experts.
- Explain mahalanobis distance.
- The referrals to Fig. S4 in the main text are not correct.
- Fig. 6G there is no statistics on the correlation analysis.
- For next revision, please add page numbers and or line numbers for easier referral to the text.

Reviewer #2

(Remarks to the Author)

Reviewer Comments for Manuscript NCOMMS-25-11127:

The manuscript titled "A deep single-cell mass cytometry approach to capture canonical and noncanonical cell cycle states" by Amouzgar et al. describes an innovative CyTOF methodology employing a 48-marker panel for precise single-cell characterization of both canonical and noncanonical cell cycle states. The proposed methodology overcomes limitations of conventional approaches by providing extensive resolution of cell cycle dynamics and differential pharmacological responses across multiple cell types, including primary human T cells.

This work provides substantial technological and biological advancements, presenting a valuable tool for exploring intricate cell cycle dynamics. The identification and characterization of noncanonical cell cycle states and cellular responses to pharmacological interventions are noteworthy contributions. The study's methodological rigor is commendable, and the conclusions drawn are robustly supported by the presented data. I recommend acceptance for publication in Nature Communications, after clearly addressing the following comments:

1. Clarification and Definition of Canonical vs. Noncanonical Cell Cycle States: Clearly articulate comprehensive definitions and distinguishing features of canonical and noncanonical cell cycle states, ideally within the Introduction section. Given the interdisciplinary audience of Nature Communications, clearly delineating the canonical states and examples of noncanonical states will significantly enhance readability and interpretability.
2. Detailed Justification for CyTOF Marker Selection: Provide explicit rationale supported by literature for the selection of each specific marker within the 'minimal', 'core', and 'complete' CyTOF panels. Clearly outline how each chosen marker facilitates achieving the objectives and intended analytical depth of each panel. This addition will significantly improve methodological transparency and replicability.
3. Enhanced Interpretation and Biological Context of FlowSOM Clusters: Figures 3E-F present intriguing FlowSOM cluster-based distinctions among various cell lines. However, the biological interpretation and cell cycle attributes associated with each cluster remain insufficiently described. Provide detailed explanations of the cell cycle characteristics and biological relevance attributed to each cluster to substantially clarify the presented data.
4. Adherence to Consistent Reference Formatting: Conduct a thorough review of all manuscript references to ensure compliance with the formatting standards of Nature Communications. Consistency in reference styling throughout the manuscript should be maintained rigorously.
5. Explicit Inclusion of IRB Approval Details: Clearly specify the Institutional Review Board (IRB) approval number or equivalent ethical oversight information in the Methods section, emphasizing ethical transparency in studies involving primary human cells.
6. Optimization of Figure Presentation for Enhanced Readability: Improve the readability and visual clarity of figures (e.g., Figures 1B, 1C, 1D, 2A, 2B, 3A, 3G) by adjusting figure resolution, sizing, and labeling. Ensuring optimal legibility will significantly enhance reader comprehension and the manuscript's overall visual impact.

Reviewer #3

(Remarks to the Author)

Version 1:

Reviewer comments:

Reviewer #1

(Remarks to the Author)

I thank the authors for the thorough revision of their manuscript. The additional dataset, the improvements in analysis, and the expanded explanations throughout the point-by-point address my comments in a convincing way.

The inclusion of 3.5 million additional single-cell measurements across replicates and drug treatments significantly improves the strength of the conclusions. I appreciate that the data have been made fully accessible to the community along with example scripts. This will likely make the platform more broadly usable.

Regarding batch effects and the treatment of replicates, I understand the authors' decision to move forward primarily with the larger dataset, and I accept this choice. Although the similarity between replicates could have been better, the steps taken to evaluate and minimize experimental variation are appropriate.

For the suggestion on mutation analysis, I initially had in mind well-known driver mutations (e.g., TP53, PTEN, NOTCH1) that could influence phenotype. The authors instead chose to perform a comprehensive SNV and CNV analysis of over a thousand cell cycle-related genes. This goes beyond what I suggested, and I appreciate the effort to integrate these data with the observed protein-level phenotypes.

The additional drug replicates, and the more detailed statistical treatment of marker variance and abundance, make the findings more believable. The use of mixed modeling, variance partitioning, and differential comparisons between conditions was particularly helpful in clarifying marker-level contributions across cell lines.

Thank you also for attempting the trajectory inference, despite it not yielding meaningful results. I agree with the assessment that single time points following perturbation make interpretation difficult, and I appreciate the authors being transparent about the limitations of these methods in this context.

I also reviewed the accompanying R scripts. The main analysis pipeline is well-structured and covers the key steps described in the manuscript. The aberrancy scoring function is technically solid and flexible, but could benefit from a small runnable example in the tutorial. But this is just a suggestion for usability and it is not required for paper acceptance. Overall the scripts are of high quality and suitable for community use.

In summary, I am satisfied with the revisions and support publication of this manuscript in its current form.

(Remarks on code availability)

I also reviewed the accompanying R scripts. The main analysis pipeline is well-structured and covers the key steps described in the manuscript. The aberrancy scoring function is technically solid and flexible, but could benefit from a small runnable example in the tutorial. But this is just a suggestion for usability and it is not required for paper acceptance. Overall the scripts are of high quality and suitable for community use.

Reviewer #2

(Remarks to the Author)

The authors addressed the reviewer's comments in an appropriate way and the manuscript is now ready for publication

(Remarks on code availability)

The current R code is very clear regarding the overall workflow and required R packages. However, since the core MELD scoring step depends on a separate Python package (meld) and a working conda environment, it would greatly improve reproducibility and accessibility for other researchers if instructions for setting up the Python/conda environment and installing meld were included in the README file, rather than only in the code itself.

The official documentation (for reticulate and meld installation) is available I recommend simply adding a direct link and a brief note in the README to guide users. This will help ensure users can easily access the most current installation instructions and minimize potential confusion.

Reviewer #3

(Remarks to the Author)

(Remarks on code availability)

Introductory remarks to Reviewers:

Thank you to the reviewers for their useful comments and suggestions. We have included substantial new single-cell data using our scMC CC platform as well as published variant profile information to address all suggestions made. Our revisions include three additional supplemental figures and some minor shuffling of previous and new supplementals to strengthen the narrative. Main **Figure 3** has seen minor changes with the suggestion of moving a figure to the supplement, and main **Figure 5** now includes an integrative modeling strategy of this new scMC data. **Figures 1, 2, 4, and 6** have minor updates to reflect smaller changes such as aesthetic improvements to font size or visualization updates but the core data and results remain untouched. We have added a new **Figure S1** to help clarify our scMC panel design, and included many supplementary plots to address reviewer comments—these plots are reported in the point-by-point when relevant. Below, we have directly addressed your comments in a point-by-point, and will call to figures, supplements, and page numbers as assigned in this revised manuscript. For convenience, we have explicitly specified and included the exact changes to the manuscript text and figures for reference, along with any additional explanations for clarity. Reviewer comments are organized as a numbered list with *Reviewer#.point #* (eg, R#1.1) and will be referenced as such. In the revised main manuscript, we have included a tracked change version of text or commented on figures that are updated. All data is part of a DOI-linked digital repository and has been made fully publicly available. Anyone can now access the data on Zenodo at [10.5281/zenodo.14852933](https://doi.org/10.5281/zenodo.14852933).

Legend:

Black text=author responses

Blue text=original reviewer comments

Figure PBP#=Figures included in the Point-By-Point for additional clarity

REVIEWER COMMENTS

Reviewer #1 (Remarks to the Author):

R#1.1. In the manuscript by Amouzgar et al., the authors have developed a very detailed CyTOF panel for cell cycle analysis. This is a great addition to the field, and interestingly the authors demonstrate that it can be used to classify cells into canonical and noncanonical cell states. The manuscript is well-written, and the figures are generally well-structured. The analysis takes a broad, global approach by examining all markers simultaneously, which provides a holistic overview. **However, currently it is difficult to tell which specific cell cycle-related markers are most informative, particularly in relation to cell line heterogeneity, drug sensitivity, and distinctions between canonical and noncanonical cell cycle states. I would thus recommend that the**

authors refine the analysis to highlight key marker differences by using additional statistical comparisons.

We agree that there is significant value in adding additional analysis to quantify and understand the relationship across biological factors like cell line heterogeneity, drug sensitivity, cell cycle aberrancy, and cell cycle phase. In addition to their individual contributions to changes in cell cycle-related markers, these multiple layers of biology can interact with each other to have differential impacts on cell cycle state. As the reviewer suggests, cell lines can have different drug sensitivities, indicating a two-way interaction. Another potential interaction could be different cell cycle aberrancies induced by a drug based on cell line identity, indicating a three-way interaction. Statistically deconvolving these behaviors requires a complex design matrix of data that includes measurements from all cell lines, drugs performed in each cell line, cell cycle phase annotations, and aberrancy scores to identify canonical and noncanonical cells. In our initial submission with the five cell lines, Jurkat cells were the only cell line treated with cell cycle pharmacologic inhibition using Palbociclib, Hydroxyurea, and Nocodazole.

To robustly address the complex quantitative and statistical design of the problem, we generated new data consisting of 36 samples and 3,544,481 single-cell measurements of all cell lines with replicate controls, and treated with all 3 drugs for ~16 hours using a subset of our scMC platform measuring cell cycle related molecules, transcription factors, DNA synthesis and content intercalators, *de novo* protein synthesis, phospho-proteins, and more. We include this data in three analysis strategies throughout the paper to support the existing results. This 3.5 million cell dataset excludes apoptotic cells (cPARP+), though we provide the raw data for these cells as well. Briefly, we **(1)** include statistical comparisons using this new data to identify single factor differences within biological factors (eg, cell lines). **(2)** We use mixed modeling to partition variance of all measurements for main effects as well as two-way, three-way, and four-way interaction effects across cell line, cell cycle phase, drug treatment, and cell cycle aberrancy. **(3)** We include an extensive differential abundance analysis using generalized linear models for pairwise statistical comparisons multiplexed with these different biological factors in tandem and provide these results in a supplementary table, which can serve as an additional resource. In addition to insights gained from our own analysis, this data is a resource for scientists to more deeply probe these layers of biological information with cell cycle state, as well as a useful dataset for the development or benchmarking of new single-cell methods that use biological or perturbative information. We include the new debarcoded, normalized, anchor-adjusted fcs files in our Zenodo repository and provide a reduced single-cell csv file equally sampled for all cell lines, replicates, and conditions to simplify data accessibility. We use this new dataset to address many questions asked by the reviewers throughout the point-by-point.

Results and manuscript edits:

Before diving deep into answering these questions, we would like to first introduce and summarize the new data generated consisting of 36 samples and 3,544,481 single-cell measurements of NALM6, U937, HEL, 293T, and JURKAT cell lines with NoTx, Palbociclib, Hydroxyurea, and Nocodazole conditions.

Cell line replicates that were seeded in new wells and cultured separately for ~48 hours before fixation were robustly consistent in each cell line. Cell cycle phase fractions were similar

across replicates (**Figure PBP1A**). Median cell cycle molecule abundance was also consistent in each cell line, with replicates for each cell line tightly hierarchically clustered (**Figure PBP1B**). We include additional univariate statistical comparisons comparing all pairwise cell lines to each other (**Figure PBP1C**). As previously reported, JURKAT and NALM6 cell lines were generally more similar to each other, with fewer significant differences in their CC-related molecules compared to other cell lines, reflecting their shared lymphoblastic leukemic origins (**Figure PBP1A-C**). Differences between JURKAT and NALM6 mostly included transcription factors TCF1, RUNX1, and FoxM1, which are known to regulate cell lineages in addition to cell cycle. Similar to before, 293T and U937 were generally more distinct from other cell lines, partially owing to their epithelial-like and myeloid-like cell origins that are distinct from the lymphoblastic JURKAT/NALM6, and erythroleukemic HEL cell lines. Differential abundance results between cell lines are provided in **Table S6**. Revised **Figure S1** also includes a rank list of CC markers with weighted utility for CC phase across cell lineages.

Figure PBP1: included in Manuscript as **Figure S4A-C**

Figure S4: Canonical and noncanonical CC states across cell lines
(A) CC fractions across different cell line replicates. **(B)** Hierarchically clustered heatmap of mean cell cycle molecular abundance across cell line replicates. **(C)** Differential abundance analysis of all pairwise cell lines for different cell cycle related markers with # of significant features (barplot), scaled model coefficient estimate aka change in abundance (heatmap), and significance labels defined by $p_{adj} \leq 0.1$ and estimates ≥ 0.25 (points).

Next, as JURKAT cells were the primary cell line analyzed in our original manuscript, we evaluated multiple replicates of each drug treatment. We cultured three JURKAT replicates for ~18 hr treatment using Palbociclib, Hydroxyurea, and Nocodazole and removed pre-apoptotic cells (**Figure PBP2A**). Cell cycle phase fractions for replicates of each treatment were consistent (**Figure PBP2B**). Median abundance of cell cycle molecular abundance was also largely consistent, with replicates from the same treatments hierarchically clustering together (**Figure PBP2C**).

Figure PBP2: included in Manuscript as Figure S6A-C

Figure S6: Drug perturbation effects

(A) Removal of cPARP+ pre-apoptotic cells. # of cells before gating; WT: 390074, PALBO: 232619, NOC: 97727, HU: 295362. Remaining normal cells after removal: WT: 386,643, PALBO: 229,629, HU: 263,428, NOC: 53,129. **(B)** Cell cycle fractions for 3 replicates of each drug in Jurkat cells. CC embeddings faceted by treatment condition. **(C)** Heatmap of molecular abundance for all replicates.

To be thorough, we checked batch effects between the prior experiment and this new, larger experiment (**Figure PBP3A-B**). We observed large experimental effects but batch effects were largely reduced by simply equally sampling jurkat treatment conditions followed by mean centering and scaling each dataset individually using the NoTx condition (train set) and extending the respective normalization parameters to the other conditions (test set) (**Figure PBP3B**). Using this strategy, we observe that the replicates from this new experiment were very similar to those of the prior experiment in both their hierarchical clustering of each mean expression across each sample as well as the mixing of cells from each dataset in the UMAP embedding (**Figure PBP3C**). Jurkat drug treatment induced aberrancy was also increased when breaking down by replicates (**Figure PBP3D**). In addition to mean centering and scaling relative to the untreated condition, we attempted deeper unsupervised batch effect correction using harmony. However, both correction strategies reduced cellular diversity (**Figure PBP3E**). This poses a problem in capturing cellular diversity across drug treatments or cell lines. In

summary, we believe drug treatments from this new dataset are in agreement with the older data procured in the original manuscript with the bonus of including substantial replicate samples and drug treatments in each cell line, all in one larger dataset free of experimental effects. With the robust similarities between the prior and new datasets, but the notable loss of feature information through unsupervised integration, we proceed to answer questions by analyzing this larger dataset.

Figure PBP3:

(A-B) UMAP embeddings of Jurkat cells from NoTx, PALBO, HU, and NOC conditions with shared cell cycle features of prior experiment and newly curated experiment: **(A)** includes before batch correction and **(B)** after batch correction performed by mean centering and scaling the datasets individually using the NoTx conditions as the training set. **(C)** Hierarchically clustered median expression of shared cell cycle features after mean centering and scaling old and new Jurkat treatment experiments. **(D)** Distribution of CC distance to nearest 5 WT neighbors grouped by each replicate across 3 experiments. **(E)** Cellular diversity (defined by pairwise euclidean distance) for each sample before correction, after correction with mean centering and scaling, and after correction with harmony.

Observing robust cell cycle state consistency across cell lines and drug treatment replicates, we next address the reviewer's request for additional analysis to better deconvolve the effects of cell line, cell cycle, cell aberrancy, and drug effects on cell cycle-related markers.

Quoted reviewer comment: "...it is difficult to tell which specific cell cycle-related markers are most informative, particularly in relation to cell line heterogeneity, drug sensitivity, and distinctions between canonical and noncanonical cell cycle states."

We address this point in a multi-faceted way. Consistent with our prior results describing differences in cell cycle-related molecules across cell lines in Manuscript **Figures 2-3**, cell lines occupy different phenotypic spaces in the embedding that persisted with CC arrest due to drug treatments (**Figure PBP4A**). Hierarchical clustering of median expression (pseudobulks) shows that even with CC arrest, cell line CC states are skewed to cluster together rather than based only on drug treatment, suggesting cell line origins being the primary source of heterogeneity (**Figure PBP4B**). JURKAT and NALM6 cell lines - which are both lymphoblastic leukemic cell lines - have more similar CC signatures to drug treatment than compared to other cell lines. Transcription factors and CC molecules like CTCF, FoxM1, Geminin, Ki67, SLBP have variable expression with drug response in different cell lines. For example, CTCF and Geminin in Jurkat cells increase with hydroxyurea treatment but U937 cells maintain lower relative expression of these markers with little change in untreated or treatment setting (**Figure PBP4C**). Ki67 expression is highly sensitive with all three drug treatments in NALM6, JURKAT, and HEL cells but 293T Ki67 expression only changes with Hydroxyurea treatment, and minimal changes in U937 cells. These statistical comparisons for drug treatment versus no treatment as well as cross-drug comparisons are included in **Table S6**. These results introduce the new dataset and qualitatively summarize that cell line identity is a major driver of variation compared to drug treatments with variable drug sensitivity across cell lines. We more deeply characterize and quantify these drug effects below.

Figure PBP4: included in Manuscript as Figure S7A-D

Figure S7: Drug perturbation effects

(A) CC embedding of cell lines from NoTx, Palbociclib, Hydroxyurea, and Nocodazole treatments. **(B)** Heatmap of all replicates for cell lines from NoTx, Palbociclib, Hydroxyurea, and Nocodazole treatments. **(C)** Heatmap of coefficient estimates (fold-change) comparing drug treatment to no treatment in each cell line. Significant features are detected with an adjusted-pvalue threshold ≤ 0.1 , and minimum absolute value coefficient (fold-change) threshold of 0.25 **(D)** Percentage of variance explained for all main and interaction effects in each feature.

To understand the effects of cell line, CC phase, drug treatment, and CC aberrancy, we use random effects modeling with main effects and interactions to partition variance explained by each feature across these different biological factors (**Figure PBP5H**). We discretize single-cells into their general CC phases (G0G1, S, G2, and M) and detect CC aberrancy using euclidean distance from untreated cells as landmarks described in the manuscript to label cells into canonical and noncanonical states. We then collectively model these categorical variables of cell line identity, drug treatment, cell cycle phase, and CC aberrancy state with main and interaction effects in each feature to interpret drivers of variation.

Variance partitioning revealed that cell line identity drives the most differences, followed by cell cycle phase, and moderate contribution due to cell cycle aberrancy or treatment effects that interact with cell line identity (**Figure PBP5I**). IdU and pH3(s10) were used to manually discretize S and M phase cells for analysis and had high variance for cell cycle gates, which may mask their contribution to other variables (**Figure PBP5I**). For all cell cycle measurements, we binned variance explained in each coefficient into <5%, 5-10%, 10-20%, and >20%. Notably many transcription factors involved in CC regulation like RUNX1, EZH2, TCF1, FoxM1, and Jarid2 were partially explained by CC aberrancy (**Figure PBP5J, PBP3D**). Cell-line dependent drug sensitivity was detected in DNM1, FoxM1, WGA, EZH2, pRb, SLBP, pCDC2, and DNA content. Molecules like SLBP, PCNA, p-CDC2, pRb, MLL1, and *de novo* protein synthesis also demonstrated cell-line dependent variance with treatment or phase. PLK1, DNMT1, and EZH2 had partially explained variance by interactions across cell line, phase, and aberrancy. This analysis demonstrates the complexity of how these different biological factors are layered to create heterogeneity in cell cycle states. It is worth noting that we survey multiple biological factors in parallel using a pseudobulk design but this strategy has limitations. For example, we are likely underpowered in detecting differences in molecules that may induce CC aberrancies with both an increase or decrease in molecule abundance. Additionally, this is a univariate strategy that does not take into account the multivariate, correlative nature of the data. Other alternative single-cell statistical strategies or analysis methods that can reveal interesting biological interactions can be employed to detect more granular biology. Despite this, we detect many interesting interactions related to drug sensitivity across cell lines, phase, and CC aberrancy. Variance partition results as well as differential abundance analysis are shared in **Table S7-8**. In summary, our platform enabled us to quantify the effects of different biological factors involved in scCC state diversity.

Figure PBP5: included in Manuscript as Figure 5H-J (Figure 5A-G remained the same as original manuscript):

(H) Cartoon schematic for single-cell variance partitioning analysis across cell lines, cell cycle phase, drug treatment, and aberrancy ($n=36$ unique samples, 3,544,481 single-cell measurements: 3 replicates each cell line, 3 replicates of each drug treatment in Jurkat cells, 1 replicate of each drug treatment in NALM6, U937, HEL, 293T cells). **(I)** Results of statistical modeling of differences of noncanonical variables defined by NN phenotypic distance to untreated cells. **(J)** Binned percentage of variance explained by each main or interaction effect for each molecular target.

Manuscript text: page 16

“Partitioning variance from cell lines, cell cycle phase, treatments, and CC aberrancy using mixed modeling

Considering our platform detects single-cell heterogeneity across cell line, cell cycle phase, CC inhibitors, and CC aberrancy, we generated a massively parallel single-cell dataset of multiple cell line replicates and drugs in each cell line to quantify the main and interaction effects across cell line identity, drug treatments, CC phase, and CC aberrancy using mixed effects modeling (**Figure 5H, S6A-C**). Variance partitioning revealed that cell line identity drives the most differences, followed by cell cycle phase, and moderate contribution due to cell cycle aberrancy or treatment effects that interact with cell line identity (**Figure 5I**). For all cell cycle measurements, we binned variance explained in each coefficient into <1%, 1-5%, 5-10%, 10-20%, and >20%. Notably many transcription factors involved in CC regulation like RUNX1, EZH2, TCF1, FoxM1, and Jarid2 were partially explained by CC aberrancy (**Figure 5J, S6B**). Cell-line dependent drug sensitivity was detected in DNMT1, FoxM1, WGA, EZH2, pRb, SLBP, pCDC2, and

DNA content. Molecules like SLBP, PCNA, p-CDC2, pRb, MLL1, and *de novo* protein synthesis also demonstrated cell-line dependent variance with treatment or phase. PLK1, DNMT1, and EZH2 had partially explained variance by interactions across cell line, phase, and aberrancy. This analysis demonstrates the complexity of how these different biological factors are layered to create heterogeneity in cell cycle states. IdU and pH3(s10) were used to manually discretize S and M phase cells for analysis and had high variance for cell cycle gates, which may mask their contribution to other variables. Notably many transcription factors involved in CC regulation like RUNX1, EZH2, TCF1, FoxM1, and Jarid2 were partially explained by CC aberrancy (**Figure 5J, S6D-E**). Molecules like SLBP, PCNA, phospho-CDC2, and *de novo* protein synthesis also demonstrated cell-line dependent variance with treatment and phase (**Figure S6B**). Variance partition results as well as differential abundance analysis are shared in **Table S7-8**. In summary, our platform enabled us to quantify the effects of different biological factors involved in scCC state diversity.”

R#1.2. Additionally, to maximize the impact of this methods paper for the community, a detailed R analysis pipeline should be provided, specifically outlining the steps for distinguishing canonical from non-canonical cell cycle cells.

In the revised manuscript we have included in our github script code that calculates multinomial Log-linear Models and multi-class ROC, detects feature-level differences between cell lines using anova, example code to partition variance using main effects and a two-way interaction using mixed models, code to compute an aberrancy score using nearest-neighbor analysis, dimensionality reduction (umap), and the raw code necessary to run single-cell perturbation scores with Meld using reticulate and conda environments. For Meld, users will have to manage their own software environment/installation for the python and R interface. Users can easily copy this script and run these functions on their own data.

https://github.com/mamouzgar/2025_cellcycle_ml We have also created Supplementary **Table 10** to specify what markers were used for the dimensionality reductions and clusters.

R#1.3. Furthermore, the manuscript appears to lack statistical comparisons and validation of technical reproducibility (e.g., technical replicates for cell lines). While I do not suggest additional experiments, further analysis of the existing data and improved documentation would strengthen the study.

To this point in the revised manuscript we've provided multiple replicates for all cell lines as well as drug treatments in Jurkat cells demonstrating robust consistency across replicates, as well as robust CC state consequences across drug treatment replicates, described in **Figures PBP1-4 (Manuscript Figure S4, S6, S7)**. Please see all the responses to Rev#1.1-1.2 above this point-by-point comment. We've also included statistical comparisons comparing expression between cell lines. These results are summarized in the comments above and differential results are shared.

Here are my additional comments:

R#1.4. The authors attempt to explain differences in cell cycle-relevant molecules based on cell size, however, they do not identify a specific trend. It would be worthwhile to investigate which mutations in cell cycle-related molecules are present in these different cell lines using publicly available data. **Does the mutation status help to explain the observed heterogeneity among cell lines?**

We agree with the reviewer that investigating whether mutational status explains the heterogeneity observed in cell lines is valuable, especially considering that the cell cycle as a molecular system is highly conserved. To address this, we extracted published variant information from DebMap¹ – a resource that genomically profiles a wide range of cancer models and includes resources from the Cancer Cell Line Encyclopedia² (CCLE). Here, we found Whole Exome Sequencing (WES) data with both single-nucleotide variant (SNV) and copy number variant (CNV) information measured using the same variant caller (Mutect2³) and CNV calling (PureCN⁴) strategies to allow fair comparison of genomic profiles between four out of five cell lines: Jurkat, HEL, U-937, and NALM-6 but not HEK293T. The closest cell line to HEK293T in the DebMAP database are HEKTE cells—which are similar to HEK293T in that they are both human embryonic kidney cells but are distinct in that HEK293T cells express the SV40 large T antigen for enhanced transfection efficiency, while HEKTE cells are engineered with a tetracycline-inducible system to control gene expression—but DebMap does not have WES (DNA) information for HEKTE. Therefore, we analyze Jurkat, HEL, U-937, and NALM-6 variant profiles but we will describe HEK(293T/TE) in relation to our results with support from literature.

We included SNV analysis from four categories based on DebMap's annotations⁵: damaging, non-conserving, conserving, and silent (further details in the citation). All variant categories were included because it may be that noncanonical CC states may be a consequence of either deleterious or non-deleterious variant of undiscovered consequences that can alter molecular regulation (eg, enhancers or transcription factor motifs). We looked at variant information for 1,115 cell cycle-related genes collected from the human Molecular Signatures Database⁶⁻⁸ (MSigDb) hallmark and kegg collections, as well as revelio⁹. We compared CC genes for cell lines across three scales of variant information: **(1)** exact SNV (most specific), **(2)** variant type in a gene regardless of specific nucleotide position (eg, a missense variant in ABR regardless of specific SNV), and **(3)** copy number variants (**Figure PBP6**).

Figure PBP6: included in Manuscript as Figure S3G-I (Figures S3A-F are excluded below):

Figure S4: Canonical and noncanonical CC states across cell lines

(G-I) Variant analysis comparing CC genes for cell lines across three scales of variant information available for Jurkat, HEL, U-937, and NALM-6 but not HEK293T in the DepMap database: (1) exact SNV (most specific), (2) variant type in a gene regardless of specific nucleotide position (eg, a missense variant in ABR regardless of specific SNV), and (3) CNVs. (G) Upset plot of exact SNVs across different cell lines. 1 unique missense SNV shared between Jurkat and NALM-6 cells: a Guanine to Adenine nucleotide change in ABR at ENST00000302538.10:c.2285G>A. (H) Binary heatmap of variant type in a gene regardless of specific nucleotide position. (I) Heatmap of genes binned for increased, no change, or decreased CNVs detected in a cell line.

For (1), we notably find that unique CC-related SNV variants of varying types (but mostly missense) were most enriched in Jurkat cells, followed by NALM-6, U-937, then HEL (Figure PBP6G). There was 1 unique missense SNV shared between Jurkat and NALM-6 cells: a Guanine to Adenine nucleotide change in ABR at ENST00000302538.10:c.2285G>A. Interestingly, Jurkat and NALM-6 are share leukemic cell line origins - Jurkat cells are CD4 T lymphocyte cells while NALM-6 cells are B cell precursors.

(2) Considering the poor concordance between exact SNVs across cell lines, we collapsed SNV types at the gene level, such that genes with variants of the same type are given the same category regardless of SNV location (Figure PBP6H). For example, missense variants in CDCA2 would be grouped as “CDCA2: missense_variant” even if the exact SNVs are different. Interestingly, leukemic cell lines Jurkat and NALM-6 cells had the most similar CC variant profile, and both shared CDCA2 variants with U-937, a monocytic cell line. HEL did not have any variants shared. Notably, these variant profiles are consistent with our scMC CC co-abundance analysis, where Jurkat and NALM-6 are most similar to each other and share some similarity with U-937. The remaining CC-related SNVs unique to each cell line are included in Table S2.

The final variant analysis in (3) CNVs tells the same story (Figure PBP6I). Jurkat and NALM-6 share CNV losses in CDKN2A and CDKN2B. While U-937 do not share CNVs with other cell lines, they do exhibit their own CNV gains and losses, while HEL cells have the

greatest number of CNV gains and losses. Interestingly, PCNA is included in our MC panel and is one of the only genes with CNV gains in U-937 cells. In our **Figure 2D** (copied below) summarizing protein variance expression across cell lines, we observe that PCNA protein abundance is highest for U-937 cells across all major CC phases, despite being the second smallest cell line using WGA as a proxy for size. Based on all these results, the CNV profiles are in agreement with results from our CC-related SNV analysis, which puts them both in agreement with our scMC study. CC-related CNV variants are included in **Table S3**.

Figure below copied and cropped from Figure 2D.

Median molecular abundance of CC targets for each cell line normalized to median across all cell lines, each equally sampled.

Notably, these SNV/CNV profiles and our scCC patterns also correlate with similarity between cell origins: Jurkat and NALM-6 cells share leukemic lymphocytic-origins, U-937 are a different branch of hematopoietic origins as a pro-monocytic cell line, and HEL cells are erythroleukemic—arguably a more distinct hematopoietic origin compared to the lymphoid and myeloid origins of the other cell lines. While HEK293T cells were unavailable, positioning their variant profile relative to others is well-supported. HEK203T cells were the most distinct from other cell lines in our scCC MC analysis, which is likely explained by a multitude of factors. First, HEK cells are immortalized human embryonic kidney cells but are not sourced from cancer origins—which would be consequential for CC aberrancy. Second, integrin-mediated cell adhesion and downstream signaling components have been shown to regulate the cell cycle, such as passage through the G1S transition and cytokinesis¹⁰. Third, there is evidence for transcriptional divergence when comparing the parental adherent HEK cells to their suspension counterparts. Specifically, transcriptional changes in metabolism was the primary difference between adherent and suspension HEK cells¹¹, and metabolic rewiring can directly regulate cell cycle proteins and cell growth via nutrient sensing signaling pathways¹². And finally, while variant profiles for HEK293T cells were unavailable, a male human embryonic kidney cell line called HA1E was available and only had two SNVs in CC-related genes: [CALD1: ENST00000361675.7:c.677T>C] and [MNX1: ENST00000252971.11:c.370G>A]. The HA1E cell line had 0 CNVs. Based on this joint literature and HEK-adjacent cell line analysis, we

suspect HEK293T cells are likely mutationally more distinct from the other cell lines, as we observed in our scCC platform.

In conclusion, variant profiles were in agreement with the cell line heterogeneity observed in our scCC MC platform, and perhaps genetic sensitivity to SNVs may have gene-level biases based on cell lineage or evolutionary origins, which is detected indirectly by our scCC panel just by measuring proteins that do not demonstrate any gene variants.

Manuscript changes summarizing these results:

We provided a detailed description of our results and conclusions in this point-by-point above but we've summarized these results in the original main text of the manuscript and **Figure S3G-I**. **Tables S2** and **S3** include variant analysis results. Text edits are on manuscript page 11 and pasted below:

Manuscript ~Page 11:

“We further investigated whether these CC differences are explained by single nucleotide variant (SNV) or copy number variant (CNV) profiles of cell cycle-related genes using DepMap¹ published information for Jurkat, HEL, U-937, and NALM-6 (293T data was unavailable). Variant profiles were in agreement with the cell line heterogeneity observed in our scCC MC platform, though the majority of gene variants do not intersect with molecules probed in our platform (except PCNA) (**Figure S3G-I**, **Table S2-3**). These results suggest that genetic sensitivity to cell cycle-related variants may have gene-level biases based on cell lineage or evolutionary origins, which are being detected indirectly by our scCC panel. While the CC is a generally conserved evolutionary program across different cell systems, we demonstrate that our approach can probe deeper into the diversity of cell states and reveal the subtle differences across similar but disparate CC molecular programs across these diverse cell lines.”

R#1.5. It would be great to have an overview cartoon figure for the function of each marker in the panel.

We have added a new **Supplementary Figure 1** that includes a network graph that maps each molecular target to a different CC-related category related to its function or identity, incorporating the fact that different molecules can have overlapping functions. A miniature version of the cartoon is included in **Figure 1B**. We've also added to our targets in **Table S1** a DOI reference and description of each molecule's cell cycle relevance. This molecular target list is also pasted in our response to reviewer #2 on bullet point R#2.2 later in this point-by-point response.

(new) Supplementary Figure 1:

Figure S1: Cartoon diagram of CC related molecules mapped to different categories
(A) Measured CC molecules are mapped to different categories. CC regulator=direct control of CC progression. PTM proteins=post translational modification. Mitotic regulator=direct control of G2M transition or mitotic progression. DNA replication, licensing, labeling, organization=measurements relating to DNA replication during S phase, DNA replication licensing, nucleotide labeling, intercalation, or histone content. Cyclins=Cyclin molecule, phosphorylated or not. Transcription factor=Any molecule also categorized as a transcription factor. Other=WGA for cell size, and puromycin for de novo protein synthesis measured using puromycin incorporation. **(B)** Feature variance across cell cycle phases to detect most variable targets for separating cell cycle progression. **(C)** Graphical diagram of features and category mapping.

R#1.6. The differences in cell cycle markers between cell lines are very interesting, and I agree with the authors' decision not to use Harmony to correct for these differences. However, I think this should rather be a technical note in the methods, as it is taking attention away from the main story line. But, I let the authors decide what they think is better.

Per the reviewers suggestion, we have moved the Harmony results to **Figure S3D-E**.

R#1.7. Statistics and number of replicates is missing from the figure legends.

We have added statistical cutoffs when relevant and replicate counts to figure legends.

R#1.8. One of the advantages of this technique is that we can find new and interesting differences between cell lines. But to do so, we need to consider technical variation of the cell lines. From the fcs file name, it seems that three replicates were pooled into one file. I would like to see an assessment of how the cell lines vary individually. Were the cell lines synchronized in each experiment?

We've provided multiple replicates for all cell lines as well as drug treatments in Jurkat cells demonstrating robust consistency across replicates, as well as robust CC state consequences across drug treatment replicates, described in response to R#1.1 *Figures PBP1-3 (Manuscript Figure S3, S5, S6)*. We've also included statistical comparisons comparing expression between cell lines (revised **Figure 3**).

We did not synchronize cell lines in our original experiments intentionally. However, our results in the revised manuscript as well as in point-by-point comments earlier show that cell lines exhibit differential drug sensitivities, and that cell synchronization induces CC aberrancies that interact with cell line identity (**Figure PBP3**). Moreover, our analyses highlight the utility of synchronizing single cell data, in general, using ground truth CC phases, like G0 OR S-phase. Therefore, we compared cell lines in an unperturbed state, and did not perform cell synchronization so that we avoid (i) comparing differential induced aberrancies, and (ii) to capture a static snapshot of all cells that captures the diverse array of cell cycle states in culture. CC synchronization (arrest) could reduce this diversity (eg, preventing cells from entering a particular phase), and looking at downstream CC states after synchronization would be confounded by other biological factors (eg, drug resistant cells proliferating faster). These are very interesting scientific questions worth pursuing in future studies. All this said, the drug treatments we've used are inherently cell synchronization, and in addition to our comparisons to the NoTx setting, we've included statistical results for all other pairwise comparisons for NoTx, Palbociclib, Hydroxyurea, and Nocodazole across different cell lines. Some of the results are summarized in **Figure S6** and also provided in **Table S7**.

R#1.9. Comparisons should also be done on a statistical basis with e.g. diffcyt or findmarkers (from scran). Likewise, what is the technical variability of the drug treatment, i.e. how does the replicates of the experiments look like.

i. "Comparisons should also be done on a statistical basis with e.g. diffcyt or findmarkers (from scran)."

All statistical comparisons previously reported as well as new analyses are made using the GLMM framework as used in diffcyt: differential abundance, unless otherwise stated (eg, variance partitioning analysis). We clarify this in the revised method section. This is complemented by the additional replications performed for this revision.

ii. "Likewise, what is the technical variability of the drug treatment, i.e. how does the replicates of the experiments look like. "

There was very low technical variability from drug treatments, as described earlier in response to R#1.1 **Figure PBP2-3. Please see our earlier response to R#1.1 of the point-by-point, also briefly summarized below.**

Briefly: JURKAT cells were the primary cell line analyzed in our original manuscript, we evaluated multiple replicates of each drug treatment. Cell cycle phase fractions for replicates of each treatment were consistent (**Figure PBP2A-B**). Additional, median abundance of cell cycle molecular abundance was also largely consistent with replicates hierarchically clustering together (**Figure PBP2C**).

Figure PBP2: included in Manuscript as Figure S6A-C

Figure S6: Drug perturbation effects

(A) Removal of cPARP+ pre-apoptotic cells. # of cells before gating; WT: 390074, PALBO: 232619, NOC: 97727, HU: 295362. Remaining normal cells after removal: WT: 386,643, PALBO: 229,629, HU: 263,428, NOC: 53,129. (B) Cell cycle fractions for 3 replicates of each drug in Jurkat cells. CC embeddings faceted by treatment condition. (C) Heatmap of molecular abundance for all replicates.

R#1.10. It would be optimal if the authors could share their R analysis pipeline for the CyTOF data, but if that is not desirable, then at least we would need information on which markers were used for generating the UMAPs and FlowSOM clusters.

We have included in our github script code that calculates multinomial Log-linear Models and multi-class ROC, detects feature-level differences between cell lines using anova, example code to partition variance using main effects and a two-way interaction using mixed models, code to compute an aberrancy score using nearest-neighbor analysis, dimensionality reduction (umap), and the raw code necessary to run single-cell perturbation scores with Meld using

reticulate and conda environments. For Meld, users will have to manage their own software environment/installation for the python and R interface. Users can easily copy this script and run these functions on their own data. https://github.com/mamouzgar/2025_cellcycle_ml We have also created **Supplementary Table 10** to specify what markers were used for the dimensionality reductions and clusters.

R#1.11. The comparison of cell lines in Fig. 3 is too holistic to understand what the main differences are. While all markers are necessary to fully distinguish the cell lines, it remains unclear which differences are most critical. Which markers are predominant in each cell line, and why? To address the “why”, I suggest examining the mutational status, as previously recommended.

We added analysis detecting CC molecular differences across cell lines in a pairwise manner **Figure PBP1 (1C included below)**. Globally, most CC-related molecules had some difference between cell lines. Molecules like CyclinB1, PCNA, Ki67, and PLK1 had moderate shifts across most pairwise comparisons (**Figure PBP1, and PBP7**). Other cell cycle regulating molecules like Rb, Geminin, SLBP, and phospho-Rb as well as WGA (cell size proxy) and CC-related transcription factors such as FoxM1 and CTCF had the largest coefficients (magnitude), indicating bigger differences. Jurkat and NALM6 cells had the fewest differences and most similar CC profiles across replicates from our correlation network analysis, reflecting their shared lymphoblastic leukemic origins.

We more deeply discuss the relationship between mutational status and cell line origins in our answer to **point R#1.4**. In summary, both single nucleotide and copy number variant (SNV, CNV) profiles were in agreement with the cell line heterogeneity observed in our scCC MC platform. The majority of genes we measured in our scCC MC platform were not mutated in these cell lines, with the only exception being increased CNVs of PCNA in U937 cells. That said, biases based on cell lineage or evolutionary origins were detected indirectly by our scCC panel regardless of the variant profiles for the specific molecules we measured. We agree with the reviewers that this is an interesting question, and that a deeper followup study using a multi-omic strategy to investigate the single-cell molecular abundance and variant profiles of different cell lines is intriguing.

Figure PBP1C: included in Manuscript as Figure S4C

Figure S4: Canonical and noncanonical CC states across cell lines
Differential abundance analysis of all pairwise cell lines for different cell cycle related markers with # of significant features (barplot), scaled model coefficient estimate aka change in abundance (heatmap), and significance labels defined by $p_{adj} \leq 0.1$ and estimates ≥ 0.25 (points).

Figure PBP7 (see next page):

Figure PBP7: Canonical and noncanonical CC states across cell lines
(A) Ridgeplots for example markers of different cell cycle control molecules as well as **(B)** protein translation, and transcription factors. **(C)** Heatmap of correlation network analysis distances for each replicate.

R#1.12. The division of cells into canonical and noncanonical states is very interesting. Could the authors try trajectory analysis (e.g. PAGA tree) or similar on the cell cycle markers to estimate at which (canonical) cell cycle stage the noncanonical cell cycle stages are generated? It would also be important to describe how exactly the cells are divided into canonical or noncanonical using example R code.

- i. "The division of cells into canonical and noncanonical states is very interesting. Could the authors try trajectory analysis (e.g. PAGA tree) or similar on the cell cycle markers to estimate at which (canonical) cell cycle stage the noncanonical cell cycle stages are generated?"*

The question about the origins of the noncanonical cell cycle states is an interesting one. Our analysis in cell lines and with drug treatments (**from main Figures 4 and 5**) suggests that noncanonical cells can exist at different phases, and it is not unreasonable to think that noncanonical cells can be potentially generated at different cell cycle stages depending on the drug treatment as well as the length or dose of the treatment. As requested by the reviewer, we performed tree-based trajectory inference and detected 3 lineages (**Figure PBP8A-C**). All lineages started from G0G1 cells and increased in expression of markers associated with CC progression like PCNA and Geminin (**Figure PBP8D**). But collectively, these lineages were not biologically sensible when looking through the lens of normal CC progression from G0G1, S, G2, then M phase (**Figure PBP8E**). Lineage1 abruptly halted in S phase. Lineage2 is IdU negative despite having molecular signatures of S phase like high SLBP and Geminin expression, and continues through a path of cells specific to HU treatment before the trajectory halts. Finally, Lineage3 is the most cyclical and similar to normal CC progression but skips S phase cells and fails to separate G2 and M cells. Lineage2 had a high density of cells specific to Hydroxyurea treatment and terminated at the end of this path (**Figure PBP8F**), and found that CC aberrancy was greater in Jurkat cells experiencing DNA replication inhibition and arrest by Hydroxyurea (**Figure PBP8G**). However, a major caveat is that none of the lineages were biologically interpretable based on our understanding of CC progression, and there are a few points to consider when designing a trajectory-based analysis strategy.

First, trajectory inference in the presence of perturbation is a known challenge¹³. Trajectory inference methods characterize dynamic biological systems in single-cell datasets by capturing linear, branching, and cyclic lineages within a system, describing cells along continuous paths. However, human disease and perturbation can corrupt normal developmental processes, resulting in perturbed trajectories that follow a similar but topologically distinct lineage from normal developmental states. The presence of perturbed cell states from disease or perturbation conditions can obscure both the discovery and the interpretability of inferred TI lineages.

Second—but related to the first point—timepoints are critical for ensemble-based algorithms like trajectory inference to have sufficient cells detect paths that will connect to terminal cell states. In our datasets, we selected a single time point where there is expected CC arrest - a terminal state. To more clearly identify a lineage that leads to aberrancy, we may need more time points between the start of treatment and CC arrest to detect the transient cells along the path, especially if they are not proceeding through the CC as expected.

Third, cell lines are transformed and are known to already have CC aberrancies, making the task of trajectory inference even more difficult when compounded with perturbations that induce CC arrest. Nonetheless, we think this is a fantastic suggestion for future followup work but is not essential for the technology and biological findings in this manuscript. We have ongoing work developing trajectory inference algorithms better suited to understand the effects of perturbation (eg, cell cycle arrest) and disease through the lens of normal progression.

Overall, it appears that trajectory methods to capture dynamic CC data still remain a challenge, but we believe the datasets provided here will serve as an ideal resource for development and benchmarking.

Figure PBP8: Trajectory analysis

Figure PBP8: (A) CC embedding of Jurkat cells colored by CC phase (B) unnormalized drug perturbation score for each condition. (C) Different trajectories for lineages detected using trajectory analysis with branching (PAGA-tree). Start node was manually selected using G0G1 cells. (D) Heatmap for key markers of CC progression. (E) Cell density in each phase for each lineage. (F) Cell density for each treatment along lineage 2. (G) Aberrancy score along lineage 2.

ii. It would also be important to describe how exactly the cells are divided into canonical or noncanonical using example R code.

We have included in our example R code functions and example data to compute the aberrancy metric to bucket cells into canonical and noncanonical groups in our github repo.

R#1.13. “Here we greatly extended existing approaches for cytometric CC analysis by capturing the expression of new proliferation molecular regulators, licensing factors, CC checkpoint inhibitors, and chromatin states”. There are indeed a lot of interesting markers in the panel, but there is almost no mention of any of them in the results or the discussion. Often when new CyTOF panels are adapted in other labs, we do not use the entire panel, but the key extra markers. So, which are the most important markers that could be used in distinguishing cell heterogeneity, drug sensitivity, and non-canonical from canonical cells?

The relevant cell cycle markers that we recommend for adaptation to other panels are those that are more directly involved in controlling cell cycle progression, which we nominate as the core panel. This includes: Ki67, Rb, pRb, CDT1, Geminin, pH3 (S10), PLK1, CyclinB1, PCNA, SLBP, IdU, and DNA. We finalized on these markers using a combination of variance with cell cycle phases and literature. In new **Figure S1B**, we've included a plot that shows variance across cell cycle phases is highest for minimal and core panel markers. Priority targets were curated based on their variance and their known functional relevance to directly controlling cell cycle progression instead of indirect control (eg, transcription factors). For example, CyclinB1 is included because it has high CC associated variance and directs progression through the G2M checkpoint. Puromycin and FoxM1 also had high variance but these molecules are excluded from the core panel because puromycin is simply a supplemental marker to detect *de novo* protein synthesis and FoxM1 is a transcription factor whose abundance more indirectly controls cell cycle progression. HH3—a marker of DNA content—also had high variance and is relevant to CC progression but DNA intercalator also had high variance but is advantageous because it is inexpensive marker included in all CyTOF panels by default and doesn't require antibody conjugation. We more clearly describe the panel rationale below.

Adapted from Figure S1:

Figure S1: (B) Feature variance across cell cycle phases to detect most variable targets for separating cell cycle progression.

We split our molecular targets into 3 categories: (i) 'minimal' includes protein and phospho-protein targets that control cell cycle checkpoints and progression, (ii) 'core' includes the minimal CC molecular targets paired with measurements of DNA content and replication such as DNA intercalators and IdU incorporation, and (iii) 'complete' includes a wide array of measured CC-related molecules including transcription factors, chromatin state, and other CC regulators that are involved in controlling CC state but not necessarily reflective of CC timing - though they can be. The core panel was selected to capture granularity in CC state across different phases of CC progression and checkpoints using a combination of published markers for monitoring CC as well as additional markers we've elected based on literature. For example, CDT1 and Geminin are part of the Fucci imaging used to monitor live cell cycle progression

and regulate DNA replication¹⁴. CDT1 abundance reflects G1 states primed to enter S phase, while Geminin peaks in late S and G2 phases to inhibit DNA replication. Ki67 and PCNA are dynamic, graded markers of proliferation that can discriminate CC phases^{15,16}. Rb and phospho-Rb control the G1/S transition¹⁷. SLBP coordinates histone protein synthesis and is tightly controlled with CC timing, peaking in S phase and rapidly degrading at S/G2¹⁸. PLK1 promotes mitotic progression and controls cell division¹⁹. Cyclins control transitions through CC checkpoints, and more specifically CyclinB1 peaks in G2 to control the G2/M transition²⁰. IdU labels active DNA synthesis, which is a defining element of S phase. CC target selection prioritizes core molecules and other essential non-CC markers (eg, cPARP, live-dead, etc), followed by inclusion of other relevant targets to the experiment that fit into remaining metal channels.

In **Figures 3-6**, we show that the majority of our targets are involved in distinguishing cell line heterogeneity, drug treatments, or CC aberrancy to varying amounts in different settings. We statistically tackle this question using variance partition analysis and differential abundance analysis using the glmm-diffcyt framework. The variance partition results are summarized in response to R#1.1, Figure PBP4:

Reviewer comment: "...it is difficult to tell which specific cell cycle-related markers are most informative, particularly in relation to cell line heterogeneity, drug sensitivity, and distinctions between canonical and noncanonical cell cycle states."

Please see our answer to R#1.1, specifically Figure PBP4. The differential abundance analysis is also included in the results with their respective figures, which is shared in parallel with the variance partition analysis from above. Finally, we've modified throughout the text and discussion to highlight specific markers, such as:

Manuscript Text, Discussion, page ~20.

"For example, Ki67—often used to identify proliferative cells—was a dynamic molecule with low expression in a subset of Jurkat cells with other molecular signatures of proliferation but also decreased across all phases in the presence of CDK4/6 inhibition. CyclinB2, but not CyclinB1, increased in G0G1 cells with all treatments, which is particularly interesting because both activate CDK1 though CyclinB2 is thought to have a less important role for viability but can compensate for CyclinB1 (57). SLBP, typically expressed in S phase, was increased in G0G1 and G2 cells treated with HU."

Minor corrections:

R#1.14. Fig. 1D, Fig. 3A, Fig. 5A, Fig. 6G text is too small.

We have modified the text size.

R#1.15. What is 1 degree T cells in Fig. 1E?

1° T cells are synonymous with primary T cells. We have swapped 1° with the word primary.

R#1.16. "cell line lineages" should be "cell lines".

We have removed "cell line lineages" .

R#1.17. Version numbers for packages used in R are missing from the methods.

Added as supplementary table #10.

R#1.18. The R script provided runs, but it could be better annotated for non-experts.
 We have improved the annotations.

R#1.19. Explain mahalanobis distance.

Mahalanobis distance is a multivariate, positive measure of each point (cell) from its population centroid, where larger magnitudes indicate further distance. We have included this in the manuscript text for **Figure 4**, where we introduce mahalanobis distance.

R#1.20. The referrals to Fig. S4 in the main text are not correct.

We have corrected the referrals in the main text.

R#1.21. Fig. 6G there is no statistics on the correlation analysis.

We have added adjusted p-value results in the figure legends. Heatmap of p-value results and Figure 6G drug correlations are included below (**Figure PBP8**). Statistics were performed at the pseudobulk level for replicates (n=3). All treatments were significantly positively or negatively correlated (padj<=0.05) except for Seliciclib.

Figure PBP8: Fig6G (left) and correlation significance for padj<=0.05 indicated by black tiles (right)

Figure PBP8: (left) Figure 6G: Correlation between raw drug treatment scores (padj<=0.05 for all correlations except Seliciclib). (right) Adjusted p-values <=0.05 (black) and >0.05 (white).

R#1.22. For next revision, please add page numbers and or line numbers for easier referral to the text.

We have included page numbers in both the revised manuscript and point-by-point.

Reviewer #2 (Remarks to the Author):

Reviewer Comments for Manuscript NCOMMS-25-11127:

The manuscript titled "A deep single-cell mass cytometry approach to capture canonical and noncanonical cell cycle states" by Amouzgar et al. describes an innovative CyTOF methodology employing a 48-marker panel for precise single-cell characterization of both canonical and noncanonical cell cycle states. The proposed methodology overcomes limitations of conventional approaches by providing extensive resolution of cell cycle dynamics and differential pharmacological responses across multiple cell types, including primary human T cells.

This work provides substantial technological and biological advancements, presenting a valuable tool for exploring intricate cell cycle dynamics. The identification and characterization of noncanonical cell cycle states and cellular responses to pharmacological interventions are noteworthy contributions. The study's methodological rigor is commendable, and the conclusions drawn are robustly supported by the presented data. I recommend acceptance for publication in Nature Communications, after clearly addressing the following comments:

R#2.1. Clarification and Definition of Canonical vs. Noncanonical Cell Cycle States: Clearly articulate comprehensive definitions and distinguishing features of canonical and noncanonical cell cycle states, ideally within the Introduction section. Given the interdisciplinary audience of Nature Communications, clearly delineating the canonical states and examples of noncanonical states will significantly enhance readability and interpretability.

We appreciate the Reviewer's positive comments about the manuscript. We have introduced canonical and noncanonical states in the introduction and clarified them in the main text as follows:

Introduction:

~Manuscript Page 2

"The global dynamics of molecules regulating CC are often conserved across different cell systems but the exact abundance and patterning of molecules required for cell cycle progression can differ depending on factors like cell size, genome size, replication speed, cell line origins, and cell extrinsic molecular factors²¹⁻²³. Furthermore, disease and perturbation can disrupt the molecular patterns that define canonical CC, inducing noncanonical CC states like CyclinD1 loss in G2 phase, relicensing of DNA replication by CDT1 overexpression during G2 phase, and tetraploidy²⁴⁻²⁷. High-throughput low-dimensional strategies may fail to capture or deeply characterize both canonical and noncanonical CC states."

Figure 4:

~Manuscript Page 12-13

“We demonstrated that deeper CC probing using our approach revealed the diversity of CC states across cell lines. CC states are often described by canonical rules like Ki67 expression as a marker for proliferative cells in S, G2, or M phase, and CDT1 licensing for DNA replication during G1. However, CC aberrancies are a hallmark of diseases like cancer, and these noncanonical CC states are reported across diverse, transformed cell lines²⁸. Examples of noncanonical CC states induced by CC perturbation can include CDT1 overexpression in G2 phase leading to relicensing of DNA replication if unregulated by Geminin, failure to degrade SLBP during G2 leading to genotoxic stress, chromosome instability from various consequences such as loss of PLK1 during mitosis leading to mitotic slippage and cellular senescence, Ki67 loss or dephosphorylation of Rb while cycling, and other mechanisms that disrupt CC progression. Noncanonical CC states have been reported in cells experiencing DNA damage, mitotic infidelity, or disruption of CC regulators^{15,25–27,29,30}. We observe that our scMC approach captures these noncanonical CC states without perturbation, such as low Ki67 abundance in S phase cells actively replicating DNA, CDT1 expression during G2, high pRb (S780) or PLK1 expression during G0G1, low pRb (S780) in G2, and additional noncanonical cell states (**Figure S2**)³¹. To label noncanonical CC states, we discretized cells into canonical and noncanonical groups based on rules for canonical CC states found in the literature for features in our core panel and observed variable fractions of noncanonical CC states between cell lines and features (**Table S4**). For example, canonical CC states with clear proliferative signatures are expected to express Ki67, or DNA licensing (CDT1) should not be abundant after S-phase, and thus deviancy from these cell states could indicate CC aberrancy. All cell lines had a subset of cells actively replicating DNA (IdU+) while Ki67 low, CDT1-expressing cells in G2, and G2 cells that were pRb (S780) low, particularly in 293T cells (**Figure S5A-C**). Manually discretized noncanonical CC phenotypes were on average 23.9% of cells (NALM6, 14.6%; U937, 16.6%; HEL, 26.5%; 293T, 36.7%, and JURKAT, 25.3%) (**Figure 4A**).”

R#2.2. Detailed Justification for CyTOF Marker Selection: Provide explicit rationale supported by literature for the selection of each specific marker within the 'minimal', 'core', and 'complete' CyTOF panels. Clearly outline how each chosen marker facilitates achieving the objectives and intended analytical depth of each panel. This addition will significantly improve methodological transparency and replicability.

We've added DOI and rationale for every molecule included to our antibody spreadsheet in revised **Supplementary Table 1**:

Appended to Table S1 (there is some redundancy as the same antibody targets have been conjugated for multiple mass channels).

Target	DOI Source	Rationale
PLK1	10.1038/s41417-025-00907-7	Key mitotic kinase essential for G2/M transition, centrosome maturation, and spindle assembly.
Geminin	10.1016/S0092-8674(00)81209-X	Inhibits DNA replication licensing and is degraded during metaphase–anaphase transition.
phospho-H3 (S10)	10.1128/MCB.22.3.874-885.2002	Phosphorylation marks chromatin condensation during mitosis.
H3K18ac	10.1177/09603271209034	Associated with transcription activation during G1/S transition and DNA damage repair

Rb	10.1007/978-1-4615-1809-9_2	Controls G1/S progression by regulating E2F transcription factors.
phospho-Rb	10.1186/s12929-022-00818-x	Phosphorylation inactivates Rb, allowing S-phase entry.
SLBP	10.1128/MCB.23.5.1590-1601.2003	Regulates histone mRNA translation, peaking in S phase.
CDT1	10.3390/genes8010002	Essential for replication origin licensing during G1; inhibited after G1 to prevent rereplication.
CDT1	10.3390/genes8010002	Essential for replication origin licensing during G1; inhibited after G1 to prevent rereplication.
CyclinB1	10.1083/jcb.115.1.1	Accumulates in G2 and controls mitotic entry via CDK1 activation.
Ki67	10.1002/(SIC1)1097-4652(200001)182:3<311::AID-JCP1>3.0.CO;2-9	Expressed in proliferating cells (G1–M) but absent in quiescent (G0) cells.
PCNA	10.1016/0014-4827(86)90520-3	Peaks in late G1/S, acts as DNA polymerase δ clamp—marker of replication machinery and S-phase progression.
PCNA	10.1016/0014-4827(86)90520-3	Peaks in late G1/S, acts as DNA polymerase δ clamp—marker of replication machinery and S-phase progression.
cPARP	10.1186/1478-811X-8-31	marker of pre-apoptotic cells
CLK1	10.7554/eLife.10288	CDC-like kinase involved in pre-mRNA splicing; links to cell cycle by regulating expression of cell-cycle genes.
HH3	10.1128/MCB.00980-09	Histone H3 total levels provide loading control; not cell-cycle regulated directly.
EZH2	10.1038/s41598-024-64338-4	positively regulates the expression of SKP2, a critical protein involved in cell cycle progression. Involved in gene repression.
p-CDC2	10.1002/j.1460-2075.1991.tb04895.x	Inhibitory phosphorylation marks G2/M checkpoint engagement.
pCyclinD1	10.1074/jbc.M113.466433	Phosphorylation regulates CyclinD1 stability—controls the transition from G1 to S phase in the cell cycle,
puromycin	10.1038/s41467-019-09128-7	puromycin incorporation is used to detect de novo protein synthesis, indicating active translation
FoxM1	10.1016/B978-0-12-407173-5.00004-2	Key CC transcription factor for G2/M genes, elevated in proliferation and mitotic entry.
pRb	10.1186/s12929-022-00818-x	Phosphorylated Rb controls CC and regulates G1/S transitio. Releases E2F, enabling S-phase gene transcription.
CyclinA	10.1083/jcb.115.1.1	Essential for S phase and G2 progression; activates CDK2 and CDK1 during DNA replication and mitosis.
cCasp3	10.1038/sj.cdd.4400476	Cleaved caspase-3 as a hallmark of apoptosis, used to assess cell death in response to cell cycle or DNA damage.

CyclinD1	10.1074/jbc.M113.466433	Activates the cyclin-dependent kinases CDK4 and CDK6 in G1 and thereby promote the cell's commitment to enter S phase.
SKP2	10.1038/nature02381	Promotes S-phase entry by targeting CDK inhibitors (e.g., p27) for degradation.
CyclinB2	10.1091/mbc.e10-05-0393	Functions similarly to Cyclin B1; involved in G2/M progression, but with different subcellular localization. Evidence suggests that it can compensate for CyclinB1 loss
CDC25A	10.2174/187152012800617678	Activates CDK2 and CDK1 via dephosphorylation, promoting S and G2/M transitions.
CDC25C	10.1186/s12935-020-01304-w	Specifically activates CDK1 to trigger mitotic entry.
CyclinD1	10.1074/jbc.M113.466433	Activates the cyclin-dependent kinases CDK4 and CDK6 in G1 and thereby promote the cell's commitment to enter S phase.
CyclinC	10.1016/s1534-5807(04)00137-6.	Regulates CDK3 and entry from quiescence (G0) into G1 phase.
CyclinE	10.1038/sj.onc.1208613	Promotes G1/S transition by activating CDK2; tightly regulated to ensure proper replication timing.
CyclinD3	10.1038/sj.onc.1202016	Activates the cyclin-dependent kinases CDK4 and CDK6 in G1 and thereby promote the cell's commitment to enter S phase.
CyclinB1	10.1083/jcb.115.1.1	Promotes mitotic entry via CDK1 activation; nuclear accumulation marks late G2.
DNA (intercalator)	10.1128/MCB.22.217459-7472.2002	DNA content dyes (e.g., DAPI, Hoechst) quantify cell cycle position by ploidy (G1, S, G2/M).
IdU	10.1038/s41467-019-09128-7	Thymidine analog incorporated during S-phase; used to label replicating DNA.
WGA	10.1002/cyto.a.23000	Cell surface lectin used to outline membranes, measure morphology, and act as a proxy of cell size
CyclinA2	10.4331/wjbc.v6.i4.346	Essential regulator of cell division cycle through the activation of kinases that participate in the regulation of S phase as well as mitotic entry (S and G2 phases); activates CDK2 and CDK1
CyclinB1	10.1083/jcb.115.1.1	Key for mitotic entry and spindle formation of G2M transition
CyclinD1	10.1016/0092-8674(94)90378-6	Activates the cyclin-dependent kinases CDK4 and CDK6 in G1 and thereby promote the cell's commitment to enter S phase.
CyclinA	10.1016/0092-8674(91)90425-K	Drives DNA replication and early mitosis via CDK activation.
CyclinB1	10.1083/jcb.115.1.1	Key for mitotic entry and spindle formation of G2M transition
CyclinD1	10.1016/0092-8674(94)90378-6	Activates the cyclin-dependent kinases CDK4 and CDK6 in G1 and thereby promote the cell's commitment to enter S phase.
p21-Waf1/Cip1	10.1016/s1368-8375(99)00049-4	Cell cycle protein that regulates and can arrest the cell cycle in G1 or S phase.

phospho-Rb	10.1186/s12929-022-00818-x	Phosphorylation inactivates Rb, allowing S-phase entry.
CTCF	10.1101/gr.241547.118	Chromatin insulator and transcription regulator, indirectly influences cell cycle gene architecture and mitotic spindle structure
RUNX1	10.1002/jcp.21738	Transcription factor regulating G1-S genes and hematopoietic proliferation.
H3K27me3	10.1016/j.molcel.2020.01.017	Repressive chromatin mark laid down by EZH2, influencing proliferation and differentiation.
Cyclin A	10.1007/978-1-4615-1809-9_9	Involved in both S phase and the G2/M transition of the cell cycle through its association with distinct cdk's
H4K16ac	10.1093/nar/gkz195	Associated with open chromatin and active transcription during S/G2.
H3K4me1	10.1371/journal.pbio.3001377	Marks enhancer regions; linked to cell-type-specific gene expression during proliferation.
MLL1	10.1038/onc.2012.352	MLL1-knockdown affects various cell cycle regulatory genes (including cyclin A, cyclin B and p57), resulting in cell cycle arrest in G2/M phase and apoptosis in cultured cells
Jarid2	10.1074/jbc.RA119.010060	Jarid2 and PRC2 regulate the cell cycle in skeletal muscle cells, affecting proliferation and mitotic exit. Regulates Rb1
CBP	10.1016/S0962-8924(97)01048-9	Transcriptional coactivator that plays an important role in a wide range of cellular processes, including proliferation, differentiation, and apoptosis. CBP controls E2F1 and CDK4 expression

R#2.3. Enhanced Interpretation and Biological Context of FlowSOM Clusters: Figures 3E-F present intriguing FlowSOM cluster-based distinctions among various cell lines. However, the biological interpretation and cell cycle attributes associated with each cluster remain insufficiently described. Provide detailed explanations of the cell cycle characteristics and biological relevance attributed to each cluster to substantially clarify the presented data.

In the revised manuscript we have added text to clarify the CC characteristics for each cluster and modified the presentation of **Figure 3** data to present the same information with more clarity and biological context.

Manuscript text: Figure 3, page 9-11

Multivariate CC quantification reveals parallel but distinct CC patterns between cell lines

...

Manuscript Page 10

Deeper probing of unharmonized clustering shows that clusters 2,8,9,10,11,and 12 capture cell lines with CC phase specificity, except for clusters 9 and 12 which have mixtures of G0G1/S or G0G1/S/G2 phases. Clusters (1,3,4,5,6,7) capture mixtures of cell lines with G0G1, S, and G2 phases defined by IdU, pH3(s10), and CyclinB1 expression. Interestingly, cluster 3 captured the majority of mitotic cells regardless of cell line status (Figure 3C-F), which is consistent with previous studies looking at multivariate molecular measurements of chromatin state across CC phases in different cell lines (38). Clusters (5,6,8,9,12) with more proliferative phases (S/G2/M) had generally greater molecular abundance of key molecules like Ki67, PCNA, and Geminin but clusters 2 is an interesting exception capturing cells with active DNA replication (IdU+) cells that are Ki67 low. CDT1 positivity in clusters 1 and 10 indicates replication licensing before S phase but cluster 10 primarily consists of HEL cells with increased abundance of transcription factors like FoxM1 and CTCF, as well as SLBP, while cluster 1 is has lower abundance of most CC-related molecules and includes diverse cell lines. High SLBP abundance in cluster 4 indicates deep progression into S phase as SLBP coordinates histone synthesis with DNA replication but again, this cluster is primarily detected in HEL and 293T cells. The majority of G2 cells are included in cluster 6, which has expected high expression of CyclinB1, PLK1, pRb (s780), and Ki67 but small fractions of G2 cells also appear in other clusters that are low in these molecules typically expressed in G2 cells, such as cluster 2. SLBP loss is observed in clusters with larger G2 fractions (6,12), which is expected since SLBP degrades rapidly during the S/G2 transition (39). In summary, cell line variance captured by the scMC CC platform is a convoluting factor in understanding cell cycle states.

Considering cell line variance confounds CC states with overclustering, we simplified the cluster task by reducing to four clusters. There were mostly pure clusters of G2 and mitotic cells in clusters 3 and 4 while G1, S, and G2 phases were mixed between clusters 1 and 2 (**Figure S3E**). These results suggest that whether under-clustering or over-clustering, basal cell line differences in CC states complicate more granular CC analysis, and that cell line variance is a strong contributor to CC state diversity. CC state integration may be necessary for experimental designs seeking to combine different systems like cell lines. But the loss in signal associated with integration, the observed purity differences between CC phases, and the mixing of CC phase and cell line identities all demonstrate that our scCC approach captures disparate CC molecular programs that are unique to each cell line.

R#2.4. Adherence to Consistent Reference Formatting: Conduct a thorough review of all manuscript references to ensure compliance with the formatting standards of Nature Communications. Consistency in reference styling throughout the manuscript should be maintained rigorously.

We have checked our manuscript references to ensure they are compliant with Nature Communications formatting standards.

R#2.5. Explicit Inclusion of IRB Approval Details: Clearly specify the Institutional Review Board (IRB) approval number or equivalent ethical oversight information in the Methods section, emphasizing ethical transparency in studies involving primary human cells.

To the manuscript methods, we have added the following statement to manuscript ~page 22:

“De-identified peripheral blood and LRS chambers samples from healthy human donors were obtained, and experiments were carried out following guidelines of the Stanford Institutional Review Board (IRB). Collections were monitored and reviewed by Stanford’s IRB. Written informed consent was obtained from all participants managed by Stanford Blood Center. PBMCs were isolated via Ficoll (GE Healthcare) density gradient centrifugation.”

R#2.6. Optimization of Figure Presentation for Enhanced Readability: Improve the readability and visual clarity of figures (e.g., Figures 1B, 1C, 1D, 2A, 2B, 3A, 3G) by adjusting figure resolution, sizing, and labeling. Ensuring optimal legibility will significantly enhance reader comprehension and the manuscript's overall visual impact.

We appreciate this feedback. We have made minor figure adjustments such as adjusted font sizes and types to meet Nature Communications guidelines (eg: Arial, minimum size= 6 in figures). These minor adjustments are unreported but any major changes to plots or new data in figures are annotated in this point-by-point response and will be included in the tracked changes of the main manuscript.

Reviewer #3 (Remarks to the Author):

Thank you for your time in reviewing this manuscript - we have found all the comments constructive and helpful.

References:

1. DepMap, B. DepMap 24Q4 Public. 30825074613 Bytes Figshare+ <https://doi.org/10.25452/FIGSHARE.PLUS.27993248.V1> (2024).
2. Barretina, J. *et al.* The Cancer Cell Line Encyclopedia enables predictive modelling of anticancer drug sensitivity. *Nature* **483**, 603–607 (2012).
3. Benjamin, D. *et al.* Calling Somatic SNVs and Indels with Mutect2. Preprint at <https://doi.org/10.1101/861054> (2019).
4. Riester, M. *et al.* PureCN: copy number calling and SNV classification using targeted short read sequencing. *Source Code Biol. Med.* **11**, 13 (2016).
5. <https://forum.depmap.org/t/what-is-the-variant-annotation-column-and-how-is-function-of-mutations-annotated/105>.
6. Subramanian, A. *et al.* Gene set enrichment analysis: A knowledge-based approach for interpreting genome-wide expression profiles. *Proc. Natl. Acad. Sci.* **102**, 15545–15550 (2005).
7. Mootha, V. K. *et al.* PGC-1 α -responsive genes involved in oxidative phosphorylation are coordinately downregulated in human diabetes. *Nat. Genet.* **34**, 267–273 (2003).
8. Liberzon, A. *et al.* Molecular signatures database (MSigDB) 3.0. *Bioinformatics* **27**, 1739–1740 (2011).
9. Schwabe, D., Formichetti, S., Junker, J. P., Falcke, M. & Rajewsky, N. The transcriptome dynamics of single cells during the cell cycle. *Mol. Syst. Biol.* **16**, e9946 (2020).
10. Kamranvar, S. A., Rani, B. & Johansson, S. Cell Cycle Regulation by Integrin-Mediated Adhesion. *Cells* **11**, 2521 (2022).
11. Malm, M. *et al.* Evolution from adherent to suspension: systems biology of HEK293 cell line development. *Sci. Rep.* **10**, 18996 (2020).
12. Diehl, F. F., Sapp, K. M. & Vander Heiden, M. G. The bidirectional relationship between

- metabolism and cell cycle control. *Trends Cell Biol.* **34**, 136–149 (2024).
13. Roux de Bézieux, H., Van den Berge, K., Street, K. & Dudoit, S. Trajectory inference across multiple conditions with condiments. *Nat. Commun.* **15**, 833 (2024).
 14. Grant, G. D., Kedziora, K. M., Limas, J. C., Cook, J. G. & Purvis, J. E. Accurate delineation of cell cycle phase transitions in living cells with PIP-FUCCI. *Cell Cycle Georget. Tex* **17**, 2496–2516 (2018).
 15. Miller, I. *et al.* Ki67 is a Graded Rather than a Binary Marker of Proliferation versus Quiescence. *Cell Rep.* **24**, 1105-1112.e5 (2018).
 16. Schönenberger, F., Deutzmann, A., Ferrando-May, E. & Merhof, D. Discrimination of cell cycle phases in PCNA-immunolabeled cells. *BMC Bioinformatics* **16**, 180 (2015).
 17. Knudsen, E. S., Buckmaster, C., Chen, T. T., Feramisco, J. R. & Wang, J. Y. Inhibition of DNA synthesis by RB: effects on G1/S transition and S-phase progression. *Genes Dev.* **12**, 2278–2292 (1998).
 18. Whitfield, M. L. *et al.* Stem-loop binding protein, the protein that binds the 3' end of histone mRNA, is cell cycle regulated by both translational and posttranslational mechanisms. *Mol. Cell. Biol.* **20**, 4188–4198 (2000).
 19. Kalous, J. & Aleshkina, D. Multiple Roles of PLK1 in Mitosis and Meiosis. *Cells* **12**, 187 (2023).
 20. Bentley, A. M., Normand, G., Hoyt, J. & King, R. W. Distinct sequence elements of cyclin B1 promote localization to chromatin, centrosomes, and kinetochores during mitosis. *Mol. Biol. Cell* **18**, 4847–4858 (2007).
 21. Uzman, A. Molecular biology of the cell (4th ed.): Alberts, B., Johnson, A., Lewis, J., Raff, M., Roberts, K., and Walter, P. *Biochem. Mol. Biol. Educ.* **31**, 212–214 (2003).
 22. Satyanarayana, A. & Kaldis, P. Mammalian cell-cycle regulation: several Cdks, numerous cyclins and diverse compensatory mechanisms. *Oncogene* **28**, 2925–2939 (2009).
 23. Zatulovskiy, E., Zhang, S., Berenson, D. F., Topacio, B. R. & Skotheim, J. M. Cell growth

- dilutes the cell cycle inhibitor Rb to trigger cell division. *Science* **369**, 466–471 (2020).
24. Stallaert, W. *et al.* The molecular architecture of cell cycle arrest. *Mol. Syst. Biol.* **18**, e11087 (2022).
 25. Zhang, H. Regulation of DNA Replication Licensing and Re-Replication by Cdt1. *Int. J. Mol. Sci.* **22**, 5195 (2021).
 26. Driscoll, D. L. *et al.* Plk1 inhibition causes post-mitotic DNA damage and senescence in a range of human tumor cell lines. *PLoS One* **9**, e111060 (2014).
 27. Shi, Q. & King, R. W. Chromosome nondisjunction yields tetraploid rather than aneuploid cells in human cell lines. *Nature* **437**, 1038–1042 (2005).
 28. Hanahan, D. Hallmarks of Cancer: New Dimensions. *Cancer Discov.* **12**, 31–46 (2022).
 29. Bruinsma, W. *et al.* Inhibition of Polo-like kinase 1 during the DNA damage response is mediated through loss of Aurora A recruitment by Bora. *Oncogene* **36**, 1840–1848 (2017).
 30. Matson, J. P. & Cook, J. G. Cell cycle proliferation decisions: the impact of single cell analyses. *FEBS J.* **284**, 362–375 (2017).
 31. Gheghiani, L., Loew, D., Lombard, B., Mansfeld, J. & Gavet, O. PLK1 Activation in Late G2 Sets Up Commitment to Mitosis. *Cell Rep.* **19**, 2060–2073 (2017).

August 2025

Introductory remarks:

We are delighted to hear reviewer satisfaction in response to our revisions, and provide our responses below.

REVIEWERS' COMMENTS

Reviewer #1 (Remarks to the Author):

1. I thank the authors for the thorough revision of their manuscript. The additional dataset, the improvements in analysis, and the expanded explanations throughout the point-by-point address my comments in a convincing way.
 - 1.1. Thank you for your comments.
2. The inclusion of 3.5 million additional single-cell measurements across replicates and drug treatments significantly improves the strength of the conclusions. I appreciate that the data have been made fully accessible to the community along with example scripts. This will likely make the platform more broadly usable.
 - 2.1. Thank you for your comments.
3. Regarding batch effects and the treatment of replicates, I understand the authors' decision to move forward primarily with the larger dataset, and I accept this choice. Although the similarity between replicates could have been better, the steps taken to evaluate and minimize experimental variation are appropriate.
 - 3.1. Thank you for your comments.
4. For the suggestion on mutation analysis, I initially had in mind well-known driver mutations (e.g., TP53, PTEN, NOTCH1) that could influence phenotype. The authors instead chose to perform a comprehensive SNV and CNV analysis of over a thousand cell cycle-related genes. This goes beyond what I suggested, and I appreciate the effort to integrate these data with the observed protein-level phenotypes.
 - 4.1. Thank you for your comments.
5. The additional drug replicates, and the more detailed statistical treatment of marker variance and abundance, make the findings more believable. The use of mixed modeling, variance partitioning, and differential comparisons between conditions was particularly helpful in clarifying marker-level contributions across cell lines.
 - 5.1. Thank you for your comments.
6. Thank you also for attempting the trajectory inference, despite it not yielding meaningful results. I agree with the assessment that single time points following

perturbation make interpretation difficult, and I appreciate the authors being transparent about the limitations of these methods in this context.

6.1. Thank you for your comments.

7. I also reviewed the accompanying R scripts. The main analysis pipeline is well-structured and covers the key steps described in the manuscript. The aberrancy scoring function is technically solid and flexible, but could benefit from a small runnable example in the tutorial. But this is just a suggestion for usability and it is not required for paper acceptance. Overall the scripts are of high quality and suitable for community use.

7.1. We have provided an example use-case running the code and added details in the README to direct users towards other software for running MELD for perturbation analysis in R using Reticulate for Python and Conda. After users have correctly installed the software, we have also provided an example starter function in the Github page to help users run MELD in R:

```
meld_using_reticulate_example_starter_function = function(df_features_only,
labels_vector){
  meld_likelihoood_score <- reticulate::import("meld")
  np <- reticulate::import("numpy")
  pd <- reticulate::import("pandas")
  pd_df = pd$pandas$DataFrame(df_features_only)
  sample_densities = meld_likelihoood_score$meld$MELD()$fit_transform(X = pd_df,
sample_labels = np$array(labels_vector))
  sample_likelihooods =
meld_likelihoood_score$meld$utils$normalize_densities(sample_densities)
  return(sample_likelihooods)
)
```

8. In summary, I am satisfied with the revisions and support publication of this manuscript in its current form.

8.1. Thank you for your review.

Reviewer #2 (Remarks to the Author):

1. The authors addressed the reviewer's comments in an appropriate way and the manuscript is now ready for publication

1.1. Thank you for your review.

Reviewer #2 (Remarks on code availability):

1. The current R code is very clear regarding the overall workflow and required R packages. However, since the core MELD scoring step depends on a separate

Python package (meld) and a working conda environment, it would greatly improve reproducibility and accessibility for other researchers if instructions for setting up the Python/conda environment and installing meld were included in the README file, rather than only in the code itself. The official documentation (for reticulate and meld installation) is available I recommend simply adding a direct link and a brief note in the README to guide users. This will help ensure users can easily access the most current installation instructions and minimize potential confusion.

- 1.1. We have added details in the README to direct users towards other software for running MELD for perturbation analysis in R using Reticulate for Python and Conda. After users have correctly installed the software, we have also provided an example starter function in the Github page to help users run MELD in R, copied below:

```
meld_using_reticulate_example_starter_function = function(df_features_only,
labels_vector){
  meld_likelihoood_score <- reticulate::import("meld")
  np <- reticulate::import("numpy")
  pd <- reticulate::import("pandas")
  pd_df = pd$pandas$DataFrame(df_features_only)
  sample_densities = meld_likelihoood_score$meld$MELD()$fit_transform(X = pd_df,
sample_labels = np$array(labels_vector))
  sample_likelihooods =
meld_likelihoood_score$meld$utils$normalize_densities(sample_densities)
  return(sample_likelihooods)
)
```

Reviewer #3 (Remarks to the Author):

2. I co-reviewed this manuscript with one of the reviewers who provided the listed reports. This is part of the Nature Communications initiative to facilitate training in peer review and to provide appropriate recognition for Early Career Researchers who co-review manuscripts.
 - 2.1. Thank you for your review.